# An end-to-end pipeline for succinic acid production at an industrially relevant scale using *Issatchenkia orientalis*

**Vinh G. Tran**[1,2,7], **Somesh Mishra** [2,3,7], **Sarang S. Bhagwat** [2,4,7], **Saman Shafaei**[1,2], **Yihui Shen**[5], **Jayne L. Allen**[4], **Benjamin A. Crosly**[1], **Shih-I Tan**[1,2], **Zia Fatma**[1,2], **Joshua D. Rabinowitz** [5], **Jeremy S. Guest** [2,4] ✉, **Vijay Singh** [2,3] ✉ & **Huimin Zhao** [1,2,6] ✉

Microbial production of succinic acid (SA) at an industrially relevant scale has been hindered by high downstream processing costs arising from neutral pH fermentation for over three decades. Here, we metabolically engineer the acid-tolerant yeast *Issatchenkia orientalis* for SA production, attaining the highest titers in sugar-based media at low pH (pH 3) in fed-batch fermentations, i.e. 109.5 g/L in minimal medium and 104.6 g/L in sugarcane juice medium. We further perform batch fermentation using sugarcane juice medium in a pilot-scale fermenter (300×) and achieve 63.1 g/L of SA, which can be directly crystallized with a yield of 64.0%. Finally, we simulate an end-to-end low-pH SA production pipeline, and techno-economic analysis and life cycle assessment indicate our process is financially viable and can reduce greenhouse gas emissions by 34–90% relative to fossil-based production processes. We expect *I. orientalis* can serve as a general industrial platform for production of organic acids.

To reduce the global reliance on fossil fuels, microbial conversion of renewable biomass into everyday products is being developed as a sustainable alternative to the conventional petroleum-based production processes[1–3]. The US Department of Energy described succinic acid (SA) as one of the top 12 bio-based building blocks that can be produced using microorganisms. It is an industrially important platform chemical with diverse applications in food, pharmaceutical, and agriculture industries[4]. SA is also used as a precursor to produce high-value chemicals, such as 1,4-butanediol and tetrahydrofuran, as well as a monomer for the synthesis of biodegradable polymers, such as polybutylene succinate[5].

Extensive research has been performed to engineer microorganisms for the production of SA. Metabolically engineered bacterial species, such as *Escherichia coli*, *Corynebacterium glutamicum*, and *Mannheimia succiniciproducens*, could produce SA with impressive performances[6–8]. Nevertheless, because of the toxicity exerted by low pH conditions on bacterial growth, utilization of neutralizing agents, such as lime ($CaCO_3$) or base (NaOH), is necessary to maintain neutral pH environment[9]. After fermentation, strong acids, such as $H_2SO_4$, are used for reacidification, converting succinate, the salt form, into SA, the undissociated form (Fig. 1A). As a result, the conventional downstream processing (DSP) steps generate a considerable amount of gypsum ($CaSO_4$), which needs proper disposal and has environmental concerns. Producing SA in low pH fermentation is an economical solution to lower operating costs and the environmental footprint. Thus, yeasts, which are more tolerant to low pH conditions, have been

[1]Department of Chemical and Biomolecular Engineering, University of Illinois at Urbana-Champaign, Urbana, IL 61801, USA. [2]Carl R. Woese Institute for Genomic Biology, University of Illinois Urbana-Champaign, Urbana, IL 61801, USA. [3]Department of Agricultural and Biological Engineering, University of Illinois at Urbana-Champaign, Urbana, IL 61801, USA. [4]Department of Civil and Environmental Engineering, University of Illinois Urbana-Champaign, Urbana, IL 61801, USA. [5]Department of Chemistry and Lewis Sigler Institute for Integrative Genomics, Princeton University, Princeton, NJ 08540, USA. [6]Departments of Chemistry, Biochemistry, and Bioengineering, University of Illinois at Urbana-Champaign, Urbana, IL 61801, USA. [7]These authors contributed equally: Vinh G. Tran, Somesh Mishra, Sarang S. Bhagwat. ✉e-mail: jsguest@illinois.edu; vsingh@illinois.edu; zhao5@illinois.edu

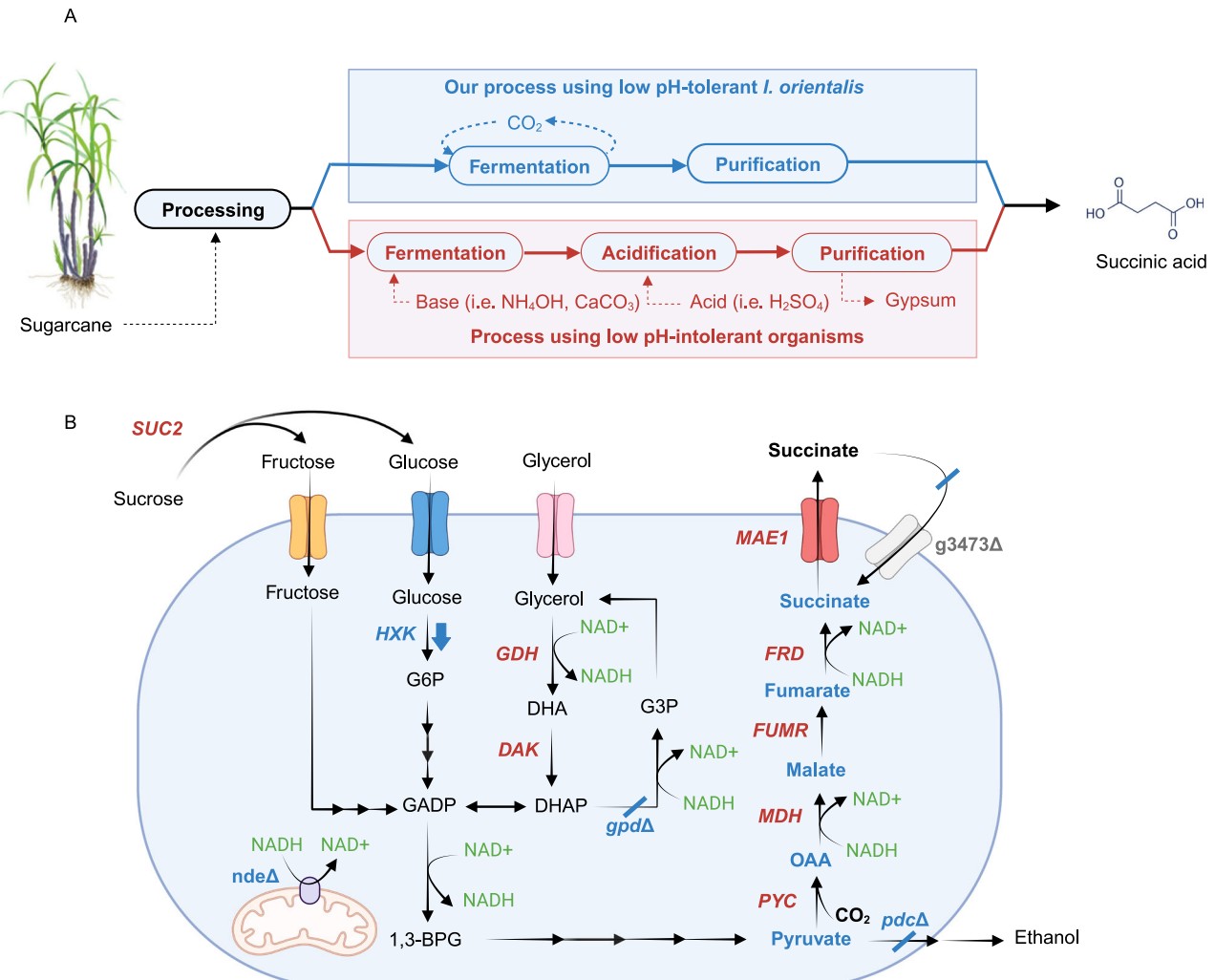

**Fig. 1 | *I. orientalis* as an industrial platform for succinic acid production. A** Our engineered *I. orientalis* strain can produce succinic acid at low pH, eliminating the requirements of base and acidification and lowering the overall production cost. The 'Sugarcane' illustration was created by Mirhee Lee, courtesy of Carl R. Woese Institute for Genomic Biology, University of Illinois Urbana-Champaign. **B** A schematic diagram for succinic acid production by engineered *I. orientalis* strain. Deleted genes are marked with delta (Δ) symbol. Overexpressed genes are marked in red. G6P glucose 6-phosphate, GADP glyceraldehyde 3-phosphate, 1,3-BPG 1,3-

bisphosphoglycerate, DHAP dihydroxyacetone phosphate, DHA dihydroxyacetone, G3P glycerol 3-phosphate, OAA oxaloacetate, HXK hexokinase, NDE external NADH dehydrogenase, GPD glycerol-3-phosphate dehydrogenase, PDC pyruvate decarboxylase, PYC pyruvate carboxylase, MDH malate dehydrogenase, FUMR fumarase, FRD fumarate reductase, MAE1 dicarboxylic acid transporter, SUC2 invertase, GDH glycerol dehydrogenase, DAK dihydroxyacetone kinase, g3473 dicarboxylic acid importer.

explored for SA production. *Saccharomyces cerevisiae* and *Yarrowia lipolytica* have been engineered to produce SA at low pH[10–12]; however, titers, yields, and productivities of engineered yeasts are lower than those achieved by bacteria.

The non-conventional yeast *Issatchenkia orientalis* is renowned for its superior tolerance to highly acidic conditions[13]. It could produce a high titer of ethanol in acidic media and was engineered to produce 135 g/L of lactic acid at low pH[14,15]. Recently, we have also developed several fundamental genetic tools to enable the genetic engineering of *I. orientalis*, including episomal plasmids, strong promoters and terminators, and CRISPR/Cas9-based system for multiplexed gene deletion[16,17]. We previously engineered *I. orientalis* SD108 to produce SA using the reductive TCA pathway (rTCA) at a titer of 11.6 g/L in batch cultures using shake flasks[13]. The pathway consisted of four enzymes: pyruvate carboxylase (PYC), malate dehydrogenase (MDH), fumarase (FUMR), and heterologous fumarate reductase (FRD) (Fig. 1B).

In this study, we report the development of metabolic engineering strategies to further improve the SA production in *I. orientalis* by

deletion of byproduct pathways, transport engineering, and expanding the substrate scope (Fig. 1B). The resulting strains produce SA at the highest titers in sugar-based media at low pH (pH 3) in bench-top reactors. We achieve a titer of 109.5 g/L, a yield of 0.63 g/g glucose equivalent, and a productivity of 0.54 g/L/h using minimal medium containing glucose and glycerol as well as a titer of 104.6 g/L, a yield of 0.63 g/g glucose equivalent, and a productivity of 1.25 g/L/h using sugarcane juice medium in fed-batch fermentations at pH 3. We also scale up our fermentation process to an industrial pilot scale in batch mode with a scaling factor of 300× and achieve SA production at a titer of 63.1 g/L, a yield of 0.50 g/g glucose equivalent, and a productivity of 0.66 g/L/h using sugarcane juice medium at pH 3. Furthermore, we perform biorefinery design, simulation, techno-economic analysis (TEA), and life cycle assessment (LCA) under uncertainty to characterize the financial viability and environmental benefits of the developed SA production pathway. Sensitivity analyses are also conducted to identify key drivers of production costs and environmental impacts for prioritization of future research, development, and deployment directions.

## Results

### Expression of a dicarboxylic acid transporter and deletion of byproduct pathways

Previous introduction of the rTCA pathway into *I. orientalis* (strain SA) (Fig. 1B) enabled the production of SA with a titer of 11.6 g/L in shake flask fermentations[13]. To further improve the titer, we attempted to express a transporter for SA. *MAE1* from *Schizosaccharomyces pombe* (*SpMAE1*) was found to be the most efficient dicarboxylic acid transporter for the export of SA[18]. The codon-optimized *SpMAE1* was integrated into the genome of strain SA, resulting in strain SA/MAE1. Strains SA and SA/MAE1 were evaluated for SA production using shake flask fermentations in minimal media (SC-URA) with 50 g/L of glucose under oxygen-limited conditions. The introduction of *SpMAE1* greatly improved the SA titer from 6.8 g/L to 24.1 g/L (Fig. 2, Supplementary Fig. 1A, and Supplementary Fig. 1B).

Ethanol was the major byproduct and accumulated at 9.5 g/L in the fermentation of strain SA/MAE1. Ethanol is formed by the reaction catalyzed by alcohol dehydrogenase (ADH), which uses NADH to reduce acetaldehyde to ethanol. Since the production of SA by the rTCA pathway requires NADH, eliminating the ethanol formation pathway might improve the SA production. Furthermore, although glycerol accumulated less than 1 g/L and was not the major byproduct observed in the fermentation of strain SA/MAE1, the glycerol formation pathway catalyzed by glycerol-3-phosphate dehydrogenase (GPD) can potentially compete with the rTCA pathway for carbon and NADH. Thus, both *PDC* (pyruvate decarboxylase) and *GPD* were deleted in strain SA/MAE1, resulting in strain SA/MAE1/pdcΔ/gpdΔ. While deletion of *PDC* to prevent ethanol formation should theoretically improve SA titer due to the increase in availability of both pyruvate and NADH, fermentation of strain SA/MAE1/pdcΔ/gpdΔ unexpectedly resulted in the similar SA titer of 24.6 g/L and the accumulation of 19.8 g/L of pyruvate under oxygen-limited conditions (Fig. 2 and Supplementary Fig. 1C).

Recently, a genome-scale model was constructed for *I. orientalis*, and all ADH activities were predicted to be localized in the mitochondrion[19]. Thus, the removal of ethanol production through the *PDC* deletion should not enhance cytosolic NADH availability, leading to no increase in SA titer. Regarding cytosolic NADH balance, glycolysis produces 2 moles of pyruvates and 2 moles of NADH from 1 mole of glucose, while the conversion of 1 mole of pyruvate to 1 mole of succinic acid requires 2 moles of NADH. Thus, although the reductive TCA cycle has the highest theoretical yield, the actual yield of SA in yeasts is limited to only 1 mol/mol glucose. We conducted [13]C metabolic flux analysis (MFA) and verified that the rTCA pathway efficiently used most of the cytosolic NADH produced by glycolysis for SA production and pyruvate excretion accounted for half of the pyruvate produced from the last step of glycolysis (Supplementary Fig. 2). We also expressed an additional copy of the rTCA pathway or individual gene of the pathway in strain SA/MAE1/pdcΔ/gpdΔ to further modulate the carbon fluxes between pyruvate and SA, but there was no significant change in SA titer or pyruvate accumulation (Supplementary Fig. 3). Thus, the shortage of NADH supply in the cytosol is the main bottleneck for SA production through the rTCA pathway.

### Co-fermentation of glucose and glycerol for SA production

Since glucose alone does not produce sufficient cytosolic NADH for SA production, other carbon sources can be considered in order to obtain higher titers and yields. Glycerol has a higher degree of reduction and thus can produce more reducing equivalents of NADH than glucose[20,21]. Since strain SA/MAE1/pdcΔ/gpdΔ exhibited growth defects in SC-URA medium with glycerol as the sole carbon source, we sought to perform fermentation of this strain using SC-URA medium

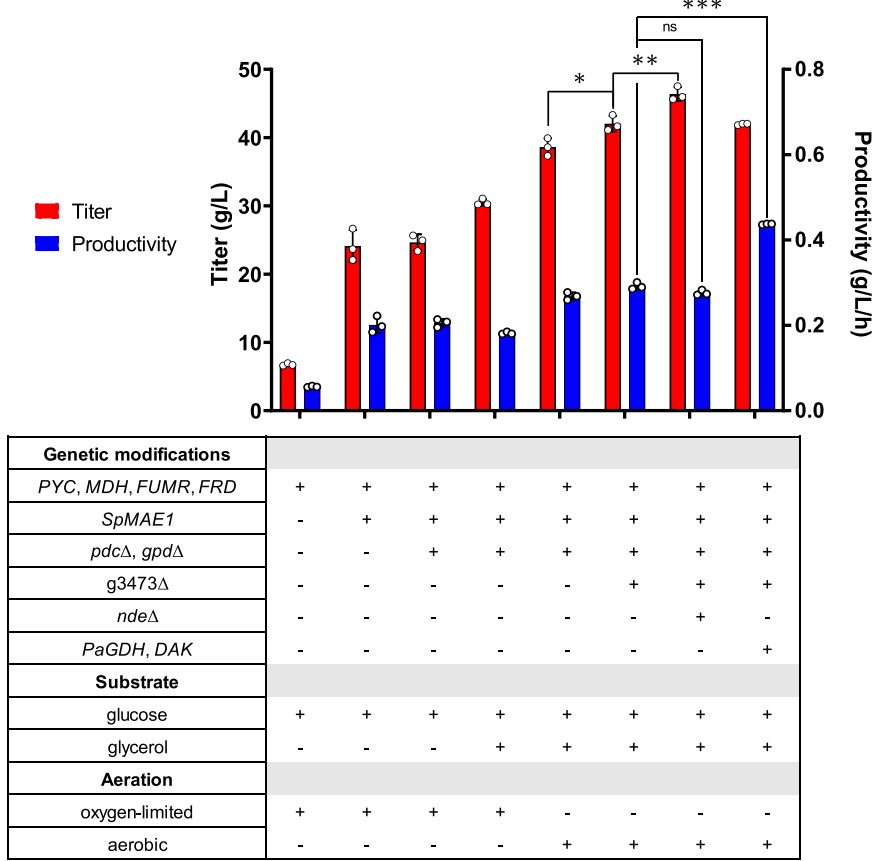

**Fig. 2 | Titers and productivities of engineered strains.** Error bars, mean ± SD ($n = 3$ biologically independent samples). *$p < 0.05$, **$p < 0.01$, ***$p < 0.001$, ns not significant, calculated by two-tailed unpaired *t*-test. Source data are provided as a Source Data file.

| Genetic modifications | | | | | | | | |
|---|---|---|---|---|---|---|---|---|
| *PYC, MDH, FUMR, FRD* | + | + | + | + | + | + | + | + |
| *SpMAE1* | - | + | + | + | + | + | + | + |
| *pdcΔ, gpdΔ* | - | - | + | + | + | + | + | + |
| *g3473Δ* | - | - | - | - | - | + | + | + |
| *ndeΔ* | - | - | - | - | - | - | + | - |
| *PaGDH, DAK* | - | - | - | - | - | - | - | + |
| **Substrate** | | | | | | | | |
| glucose | + | + | + | + | + | + | + | + |
| glycerol | - | - | - | + | + | + | + | + |
| **Aeration** | | | | | | | | |
| oxygen-limited | + | + | + | + | - | - | - | - |
| aerobic | - | - | - | - | + | + | + | + |

with 50 g/L of glucose and 20 g/L of glycerol. Previously, using glucose and glycerol as dual carbon sources was shown to enhance the conversion of oxaloacetate to malate through the increased supply of NADH from glycerol in an engineered *M. succiniciproducens*[6]. As shown in Supplementary Fig. 4A, the cells could consume both substrates for SA production under oxygen-limited conditions; however, the glycerol consumption was slow, and the SA titer was improved to only 30.5 g/L after 7 days of fermentation. We then performed the fermentation under aerobic conditions, postulating the glycerol metabolism might be limited at oxygen-limited conditions. Under aerobic conditions, both glucose and glycerol were consumed faster, allowing the production of 38.6 g/L of SA (Fig. 2 and Supplementary Fig. 4B). We also tested fermentation of strain SA/MAE1/pdcΔ/gpdΔ using SC-URA medium with 50 g/L of glucose under aerobic conditions. Interestingly, while the rTCA pathway is a fermentative pathway and higher aeration might channel more carbon flux into the TCA cycle for aerobic respiration, we observed that aerobic conditions led to similar titers compared to the oxygen-limited conditions and the cells were able to consume glucose faster, leading to higher productivities (Supplementary Fig. 4C).

## Deletions of a dicarboxylic acid importer and external NADH dehydrogenase

Further gene deletions were then attempted to increase the SA production. Recently, the JEN family carboxylate transporters PkJEN2-1 and PkJEN2-2 in *Pichia kudriavzevii* were characterized to be involved in the inward uptake of dicarboxylic acids[22,23]. PkJEN2-1 and PkJEN2-2 were annotated as g3473 and g3068 in *I. orientalis*, respectively. g3473 was deleted from strain SA/MAE1/pdcΔ/gpdΔ, leading to strain g3473Δ. Fermentation of this strain in SC-URA medium with 50 g/L of glucose and 20 g/L of glycerol improved the SA titer to 42.0 g/L (Fig. 2 and Supplementary Fig. 5A), suggesting that preventing SA from re-entering the cells was beneficial. g3068 was further knocked out in strain g3473Δ; however, we observed that disruption of both JEN2 transporters lowered the SA titer to 34.5 g/L and thus was not beneficial (Supplementary Fig. 5B). This result was inconsistent with the previous report that deletion of both JEN transporters in *P. kudriavzevii* CY902 resulted in higher SA titer than single gene deletions, which might be attributed to different genetic backgrounds. *P. kudriavzevii* CY902 was engineered to produce SA using the oxidative TCA (oTCA) pathway by deletion of the succinate dehydrogenase complex subunit gene *SDH5*, while SA production in our engineered *I. orientalis* SD108 was achieved using the rTCA pathway. Moreover, based on MFA, a small amount of cytosolic NADH was oxidized by the external mitochondrial NADH dehydrogenase (NDE), which transports electrons from cytosolic NADH to the mitochondrial electron transport chain (Supplementary Fig. 2). *NDE* was targeted for disruption in strain g3473Δ, resulting in strain g3473Δ/ndeΔ. Compared to strain g3473Δ, *NDE* deletion further improved the SA titer to 46.4 g/L, suggesting the knockout of *NDE* increased the cytosolic NADH pool for the production of SA (Fig. 2 and Supplementary Fig. 5C). Nevertheless, the disruption of *NDE* lowered the glucose consumption rate; hence, despite having higher titer, strain g3473Δ/ndeΔ had similar productivity as strain g3473Δ (Fig. 2).

## Improving glycerol consumption

The slow glycerol consumption indicated the endogenous glycerol metabolism might not be highly active. Previously, overexpression of glycerol dehydrogenase (GDH) from *Pichia angusta* and endogenous dihydroxyacetone kinase (DAK) established an NADH-producing glycerol consumption pathway in *S. cerevisiae*[24]. Thus, we sought to employ a similar strategy to improve the glycerol consumption in *I. orientalis*. The codon optimized *PaGDH* and endogenous *DAK* were overexpressed in strains g3473Δ and g3473Δ/ndeΔ, resulting in strains g3473Δ/PaGDH-DAK and g3473Δ/ndeΔ/PaGDH-DAK, respectively.

Fermentations of these strains in SC-URA medium with 50 g/L of glucose and 20 g/L of glycerol did not lead to higher titers of SA; g3473Δ/PaGDH-DAK and g3473Δ/ndeΔ/PaGDH-DAK produced SA at titers of 41.9 g/L and 46.5 g/L, respectively, similar to the titers achieved by the parent strains lacking the overexpression of *PaGDH* and *DAK* (Fig. 2, Supplementary Fig. 6A, and Supplementary Fig. 6B). However, the overexpression of *PaGDH* and *DAK* was beneficial to both glucose and glycerol utilization rates. The productivities were increased from 0.29 to 0.44 g/L/h in strain g3473Δ/PaGDH-DAK and from 0.28 to 0.32 g/L/h in strain g3473Δ/ndeΔ/PaGDH-DAK (Fig. 2 and Supplementary Fig. 6C).

Strain g3473Δ/PaGDH-DAK could produce 25.4 g/L of SA in fermentation using 50 g/L glucose, while 41.9 g/L of SA could be obtained from 50 g/L of glucose and 20 g/L of glycerol (Supplementary Fig. 7A). Since the SA titer of 41.9 g/L could also be achieved just by simply using more initial glucose in the fermentation using only glucose, one may question the advantages of using glucose and glycerol as dual carbon sources. On a carbon equivalent basis, 1 gram of glucose is equivalent to 1 gram of glycerol. Using 50 g/L of glucose and 20 g/L of glycerol enabled the SA yield of 0.60 g/g glucose equivalent, which was higher than the yield of 0.51 g/g glucose from fermentation using only 50 g/L glucose (Supplementary Fig. 7B). Furthermore, from 70 g/L of glucose, a concentration equivalent to 50 g/L of glucose and 20 g/L of glycerol, strain g3473Δ/PaGDH-DAK could produce SA at titer of only 35.6 g/L and yield of 0.50 g/g glucose (Supplementary Fig. 7). Therefore, utilizing a mixture of glucose and glycerol as carbon sources allowed SA production at higher titers and higher yields than using equivalent amount of glucose. We also compared the SA production in strain g3473Δ/PaGDH-DAK using 50 g/L of glucose and different glycerol concentrations of 10 g/L, 20 g/L, and 30 g/L (Supplementary Fig. 8). As expected, SA titer increased as the initial amount of glycerol increased; thus, a comparison between titers was not meaningful. Also, the yield was lowest when 10 g/L of glycerol were used. No significant difference in yields was observed when 20 g/L and 30 g/L of glycerol were used; however, the productivity was highest when 20 g/L of glycerol were used. Thus, we considered 20 g/L of glycerol to be the optimal concentration to use with 50 g/L of glucose for the SA production.

We also attempted to relieve the catabolite repression of glucose on the glycerol consumption through the deletion of a hexokinase, which was shown to reduce the glucose phosphorylation rate and permit the co-utilization of glucose and xylose in *S. cerevisiae*[25]. Through BLAST analysis, three potential hexokinase genes (g1398, g2945, and g3837) were determined, and only the deletion of g3837 in strain g3473Δ/PaGDH-DAK enabled the simultaneous consumption of both glucose and glycerol (Supplementary Fig. 9). While similar SA titers could be achieved, g3837 deletion lowered the glucose and glycerol consumption rates, leading to no increase in the productivity.

## Fed-batch fermentations and scale-up

Following the shake flask fermentations, we performed fed-batch fermentations to increase the titer of SA and to assess the performance of our engineered strain in large-scale production. To exploit the superior tolerance to low pH of *I. orientalis*, we chose to perform the fed-batch fermentations at pH 3. At this pH, approximately 90% of the SA species are fully protonated SA, while the remaining 10% of the species are hydrogen succinate[26]. We first tested the performance of strain g3473Δ/PaGDH-DAK, which was chosen over g3473Δ/ndeΔ/PaGDH-DAK due to higher productivity, using SC-URA medium with 50 g/L of glucose and 20 g/L of glycerol in batch fermentation in a bench-top bioreactor with a size of 0.3 L and a working volume of 0.1 L under static conditions of agitation and continuous sparging of $O_2$ and $CO_2$. We observed that the titers (27.1 g/L and 30.7 g/L at 0.333 vvm (volume per working volume per min) of $CO_2$ and 0.667 vvm of $CO_2$, respectively) were much lower than the titer obtained in shake flask fermentation (42.1 g/L) (Supplementary Fig. 10A, B). Particularly, while

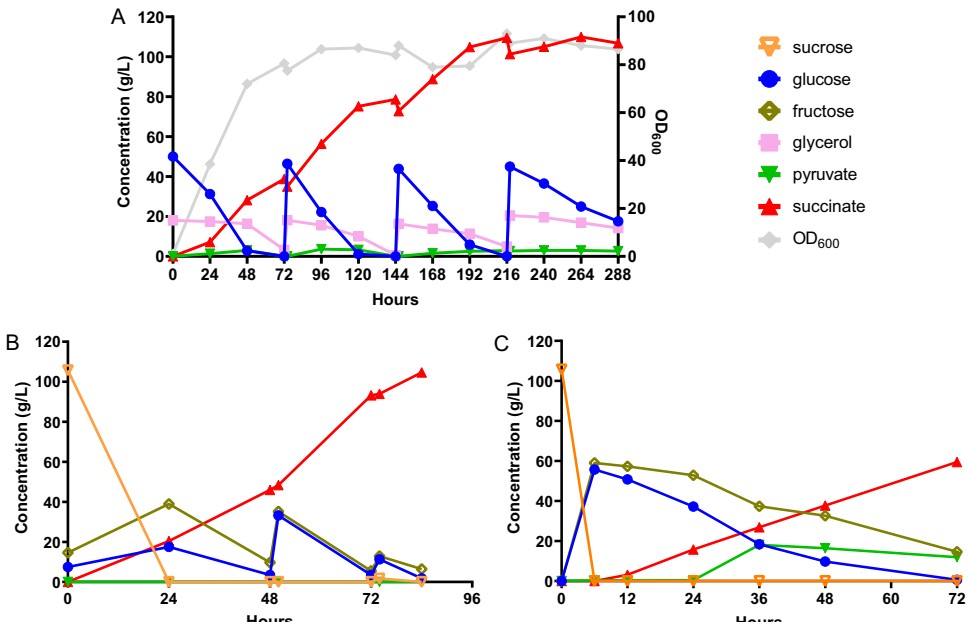

**Fig. 3 | Fermentations in bioreactors. A** Fed-batch fermentation of strain g3473Δ/PaGDH-DAK/g3873Δ in minimal medium with glucose and glycerol. **B** Fed-batch fermentation of strain g3473Δ/PaGDH-DAK/ScSUC2 in sugarcane juice medium. **C** Batch fermentation of strain g3473Δ/PaGDH-DAK/ScSUC2 in sugarcane juice medium in a pilot-scale reactor. Source data are provided as a Source Data file.

similar titers of SA could be produced from glucose in both reactor and shake flask, the SA titers produced during glycerol utilization phase were much lower in the bioreactor. We also conducted batch fermentation in bioreactor using strain g3473Δ/PaGDH-DAK/g3837Δ and observed that this strain could produce more SA during the glycerol consumption phase and a titer of 38.8 g/L of SA could be obtained at 0.167 vvm of $O_2$ and 0.667 vvm of $CO_2$ (Supplementary Fig. 10C, D). We postulated that while glycerol was being utilized in the bioreactor environment with higher aeration than the shake flask environment, more carbon flux might be channeled to the TCA cycle and led to lower SA titer; on the other hand, deletion of g3837 might repress the activity of the TCA cycle genes and improve SA production. Real-time PCR analysis was employed to compare the transcriptional levels of genes in the rTCA pathway and some selected genes in the TCA cycle (citrate synthase, *CIT*; aconitase, *ACO*; and isocitrate dehydrogenase, *IDH*) in strains g3473Δ/PaGDH-DAK with or without g3837 deletion grown in YP medium with glycerol. We observed that knockout of g3837 maintained similar expressions of genes in the rTCA pathway but lowered the expression levels of *CIT*, a homolog of *ACO*, and *IDH*s (Supplementary Fig. 11). Thus, the lower activities of genes in the TCA cycle might lead to higher SA titer obtained by strain g3473Δ/PaGDH-DAK/g3837Δ in the bioreactor. The fed-batch fermentation of strain g3473Δ/PaGDH-DAK/g3837Δ in SC-URA medium with glucose and glycerol feeding produced 109.5 g/L of SA with a yield of 0.65 g/g glucose equivalent and a productivity of 0.54 g/L/h (Fig. 3A). At the end of the fermentation, we observed the formation of crystals, which was likely SA (Supplementary Fig. 12). While other organic acids, such as lactic and acetic acids, are fully miscible in aqueous broth at pH 1–14, the solubility of SA decreases as the pH becomes more acidic[27].

Following the high fermentative performance of our recombinant *I. orientalis* strain using the minimal medium commonly used in the laboratory, we then tested the production of SA using a real industrial substrate, sugarcane juice. Sugarcane is the most energy-efficient perennial C4 plant and has a higher biomass yield compared to other crops such as switchgrass and miscanthus[28]. Furthermore, sugarcane juice, as a sucrose-based feedstock, is cheaper than glucose and starch-based substrates such as corn and cassava[29]. Since *I. orientalis* is unable

to utilize sucrose, the invertase *SUC2* from *S. cerevisiae* was expressed in g3473Δ/PaGDH-DAK. Batch fermentation occurred in the first 48 h, and SA could be produced at a titer of 46.0 g/L, a yield of 0.40 g/g glucose equivalent, and a productivity of 0.96 g/L/h. With feeding of concentrated sugarcane juice afterwards, our engineered strain could produce SA at a titer of 104.6 g/L, a yield of 0.63 g/g glucose equivalent, and a productivity of 1.25 g/L/h at the bench scale (Fig. 3B).

Furthermore, we scaled up our SA fermentation process using sugarcane juice from the bench scale to a pilot scale. Here, we determined the process parameters to maintain similar power input per unit volume and the same Reynolds number between bench-scale and pilot-scale bioreactors, and batch fermentation was performed in a pilot-scale bioreactor with a size of 75 L and a working volume of 30 L or a scale-up factor of 300× compared to the bench-top bioreactor. Our strain could produce SA at a titer of 63.1 g/L, a yield of 0.50 g/g glucose equivalent, and a productivity of 0.66 g/L/h at pH 3 (Fig. 3C). Due to the volume requirement, we did not attempt a fed-batch fermentation at the pilot scale; nevertheless, our titer and yield for the batch fermentation in the pilot-scale bioreactor were comparable to those in the bench-scale bioreactors. Thus, we anticipated a similar performance of the strain in fed-batch fermentation at the pilot scale.

We further completed the full production process of SA by devising a DSP to recover SA from sugarcane juice fermentation broth using two-stage vacuum distillation and crystallization. Without further acidification of the fermentation broth containing 63.1 g/L SA obtained from the pilot-scale fermentation, the maximum yield was 31.0% during the first stage. The filtrate from stage 1 was then concentrated to 50% of its volume using vacuum distillation and subjected to the second stage of crystallization. The yield of SA from stage 2 was 47.7%, and similar amounts of SA crystal were obtained for both stages (1.98 g for stage 1 and 2.10 g for stage 2). Thus, via two-stage vacuum distillation and crystallization, the overall SA recovery yield of 64.0% from the low-pH fermentation broth was obtained. Furthermore, the purities of SA crystals recovered in stage 1 and stage 2 were estimated to be 88.9% and 86.23%, respectively. The obtained result was in line with the findings from a previous study[30]. From the results, it was evident that the crystallization of SA in high purity (>85%) from

the untreated fermentation broth was successful. However, further investigation is needed to eliminate the coloring impurities to recover SA crystal of commercial grade.

## Techno-economic analysis and life cycle assessment

We designed and simulated end-to-end biorefineries capable of accepting sugarcane as a feedstock, saccharifying it to sugarcane juice (sucrose, glucose, and fructose), fermenting the sugars to SA using *I. orientalis*, and separating the fermentation broth to recover dried SA crystals (Supplementary Fig. 13) at an annual production capacity of 26,800 metric tonnes of SA (the global demand for SA in 2013 was approximately 76,000 metric tonnes[31]). The biorefineries were simulated under alternative fermentation scenarios with assumptions for yield, titer, and productivity corresponding to the fermentation performance achieved in laboratory-scale batch mode (laboratory batch scenario) and fed-batch mode (laboratory fed-batch scenario) experiments as well as the pilot-scale batch mode setup (pilot batch scenario). To characterize the financial viability and environmental benefits of the developed SA pathways, we performed TEA and LCA for each scenario under baseline assumptions as well as under uncertainty (2000 Monte Carlo simulations for each scenario with Latin hypercube sampling; the distribution of results at alternative numbers of Monte Carlo simulations is reported in Supplementary Table 1 and the assumed baseline values and distributions of all uncertain parameters for each scenario are reported in Supplementary Data 1). We used the minimum product selling price (MPSP, in 2016\$ with an internal rate of return of 10%), 100-year global warming potential (GWP$_{100}$; cradle-to-grave), and fossil energy consumption (FEC; cradle-to-gate) as metrics to represent the TEA and LCA results. We also performed sensitivity analyses using Spearman's rank order correlation coefficients (Spearman's ρ) to identify key drivers of production costs and environmental impacts. Finally, to set and prioritize targets for further improvements to financial viability and environmental sustainability, we designed and simulated biorefineries across the potential fermentation performance landscape (i.e., 2500 yield-titer combinations each across a range of productivities for both neutral and low-pH fermentation).

Based on the experimental performance in the laboratory batch scenario, the biorefinery could produce SA at an estimated MPSP of \$1.70/kg (baseline; Fig. 4A) with a range of \$1.51–1.92/kg (5th–95th percentiles; hereafter in brackets). The biorefinery's GWP$_{100}$ and FEC under this scenario were estimated to be 1.95 kg CO$_2$-eq./kg (1.37–2.65 kg CO$_2$-eq./kg) and -3.74 MJ/kg (−12.9–5.39 MJ/kg), respectively (Fig. 4B and Supplementary Fig. 14A). In the laboratory fed-batch scenario (with improved fermentation SA titer, yield, and productivity over that of the laboratory batch scenario), the biorefinery's MPSP was \$1.06/kg (\$0.96–1.22/kg), GWP$_{100}$ was 0.93 kg CO$_2$-eq./kg (0.71–1.32 kg CO$_2$-eq./kg), and FEC was −5.36 MJ/kg (−8.97–0.213 MJ/kg). In the pilot batch scenario (with fermentation SA yield and titer improved relative to the laboratory batch scenario but lower than those of the laboratory fed-batch scenario, and lower productivity than both laboratory scenarios), the biorefinery had an estimated MPSP of \$1.37/kg (\$1.23–1.54/kg), GWP$_{100}$ of 1.67 kg CO$_2$-eq./kg (1.22–2.17 kg CO$_2$-eq./kg), and FEC of −0.21 MJ/kg (−7.08 to 6.47 MJ/kg). A Sankey diagram depicting the flow of carbon through the biorefinery for this scenario was also shown in Supplementary Fig. 15.

Across the 28 parameters to which uncertainty was attributed for the pilot batch scenario, we found MPSP was most sensitive to fermentation SA yield (Spearman's ρ of −0.60; all uncertainty distributions are listed in Supplementary Data 1 and Spearman's ρ values for all parameters are reported in Supplementary Table 2). MPSP was also significantly sensitive to feed sugarcane unit price (Spearman's ρ of 0.39), plant uptime (−0.38), the plant's capacity for feed sugarcane (−0.31), and fermentation SA titer (−0.30). GWP$_{100}$ was most sensitive to the boiler efficiency (Spearman's ρ of −0.63). GWP$_{100}$ was also significantly sensitive to fermentation SA titer and yield, with Spearman's ρ values of −0.62 and 0.32, respectively. FEC was most sensitive to fermentation SA yield (Spearman's ρ of 0.63), and also sensitive to the boiler efficiency (−0.56) and fermentation SA titer (−0.49). To further characterize the implications of fermentation performance, we performed TEA and LCA across the fermentation performance landscape (Fig. 4C–F and Supplementary Fig. 14B, C), simulating 2500 yield-titer combinations across a range of productivities for each of two alternative regimes: low-pH fermentation (i.e., with fermentation at a pH of 3 controlled using base addition during fermentation and no acidulation required after fermentation) and neutral fermentation (i.e., with complete neutralization of SA by base addition during fermentation and complete re-acidulation after fermentation). Results for an alternative low-pH scenario with re-acidulation required after fermentation were shown in Supplementary Fig. 16.

## Discussion

Recognizing its importance as a platform chemical, researchers have spent much effort to engineer microorganisms for the production of SA from renewable biomass. In this study, we present a number of metabolic engineering strategies to improve the SA production in *I. orientalis*. We engineered the strains by deletion of byproduct pathways, transport engineering to enable SA export and limit SA import, and expansion of the substrate scope to allow utilization of glycerol and sucrose. Our final strains could produce more than 100 g/L of SA at pH 3 in fed-batch fermentations using both SC-URA medium and sugarcane juice medium. Nevertheless, lack of cytosolic NADH, as indicated by MFA, was the roadblock for higher yield. Since glycerol yields twice as much NADH compared to glucose, co-utilization of glucose and glycerol helped increase both titer and yield. Furthermore, *NDE* deletion was shown to enhance SA titer. However, *NDE* deletion also lowered substrate consumption and productivity, which might be due to its involvement in the electron transport chain and ATP synthesis. Expressing transhydrogenase to convert cytosolic NADPH produced in the pentose phosphate (PP) pathway into NADH could be another way to enhance cytosolic NADH. Nevertheless, MFA indicated the carbon flux to PP pathway and NADPH production were 10-fold less than the carbon flux to glycolysis and NADH production, respectively (Supplementary Fig. 2). Moreover, the majority of NADPH was used for threonine and lipid synthesis; thus, we considered expression of a transhydrogenase to be unlikely to significantly improve SA yield.

Comparing titer, yield, and productivity alone, bacteria are still more efficient than our engineered *I. orientalis* (Table 1). Nevertheless, SA production using bacteria is usually performed at neutral pH, increasing the expense of DSP as more than 60% of the total production cost are generated by the downstream separation and purification processes[32]. On the other hand, yeasts can tolerate highly acidic conditions and thus offer economical advantages compared to bacteria. Only engineered *Y. lipolytica* could produce SA at high titers and yields thus far (Table 1). With glycerol as the substrate, it could produce 110.7 g/L of SA with the pH dropping to 3.4 at the end of the fermentation[10]. With glucose as the substrate, *Y. lipolytica* could produce 101.4 g/L of SA, but near neutral pH of 5.5 was maintained during the fermentation[11]. So far, high SA production (>100 g/L) using engineered *Y. lipolytica* at low pH has not been demonstrated with sugar-based substrates. Furthermore, fermentations of *Y. lipolytica* were conducted in complex YP medium, while minimal media were used for SA production using our engineered *I. orientalis*. Complex media are unfavorable in industrial applications due to higher cost, and the contents of peptone and yeast extract affect the estimation of true product yield from carbon[33]. Furthermore, engineered *Y. lipolytica* employed the oxidative tricarboxylic acid (oTCA) pathway, which contained two decarboxylation steps and thus led to loss of carbon and release of greenhouse gases. On the other hand, our recombinant *I. orientalis* uses the rTCA pathway, which could fix carbon and

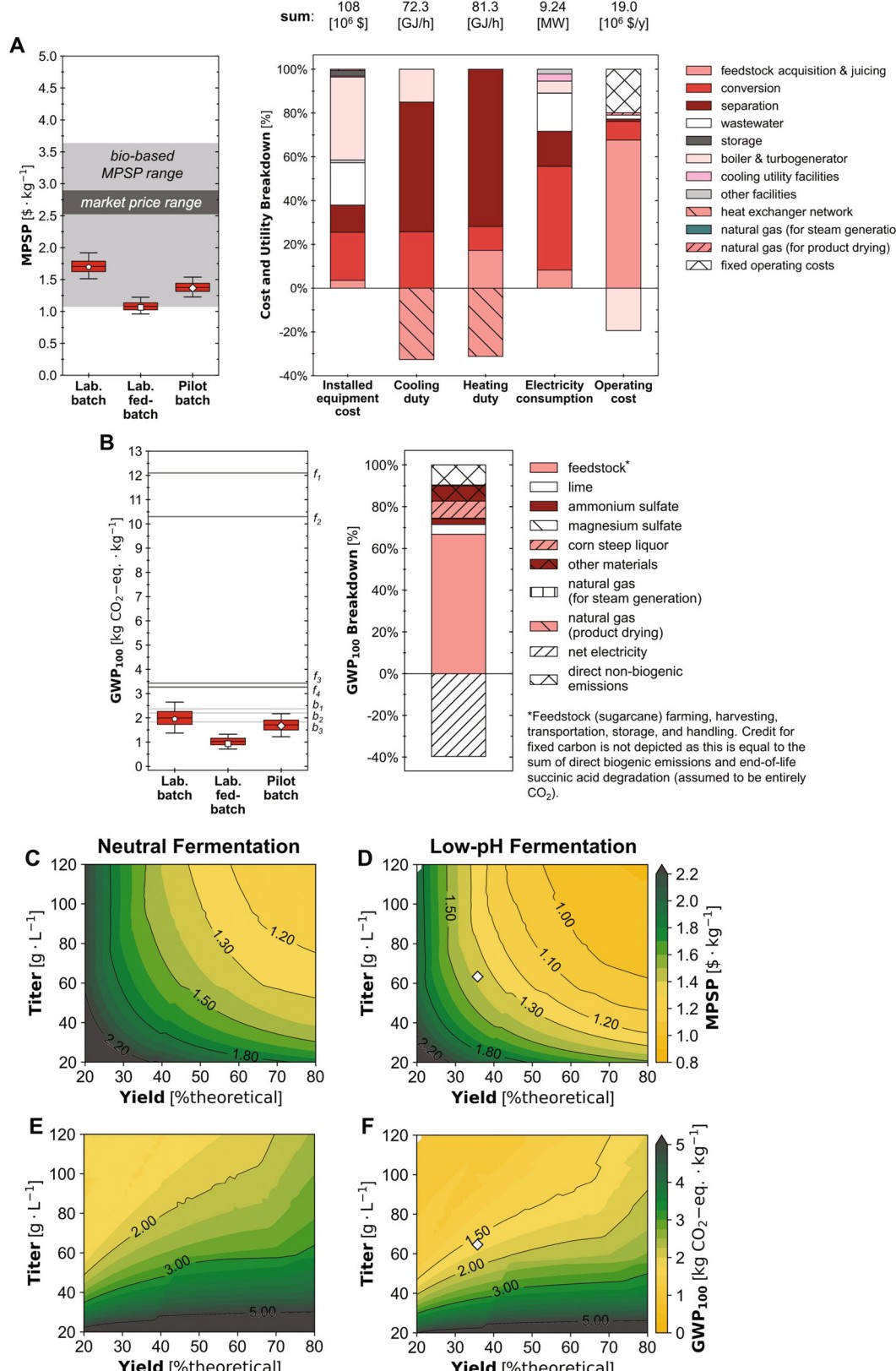

therefore could be more sustainable compared to the oTCA pathway in SA production.

Moreover, different studies have attempted to scale up the fermentative SA production[34–36]. However, almost all these studies are limited to bioreactors with total volumes less than 10 L. Low pH fermentation to produce SA via microorganisms at the bioreactor volume

of 75 L have not been attempted. Also, a few companies, such as BioAmber and Myriant, have attempted the commercialization of large-scale SA manufacturing[4,37]. At present, these projects have been abandoned, majorly due to process non-profitability[4]. The lesson learned was to include low-pH fermentation and an efficient DSP to make SA market ready[38,39]. Furthermore, several studies have

**Fig. 4 | Techno-economic analysis and life cycle assessment under uncertainty and across the fermentation design and technological landscape.** Uncertainties (box and whisker plots) and breakdowns (stacked bar charts) for (**A**) minimum product selling price (MPSP) and (**B**) cradle-to-grave 100-year global warming potential ($GWP_{100}$). Whiskers, boxes, and the middle line represent 5th/95th, 25th/75th, and 50th percentiles from 2000 Monte Carlo simulations ($n = 2000$ simulations) for each scenario. Pentagon, square, and diamond markers represent baseline results for the laboratory batch (Lab. batch) laboratory fed-batch (Lab. fed-batch), and pilot batch (Pilot batch) scenarios, respectively. Stacked bar charts report baseline results for the pilot batch scenario; results for other scenarios are included in the SI. Electricity consumption includes only the consumption of the system; production was excluded in the depicted breakdown for figure clarity. Tabulated breakdown data for material and installed equipment costs, heating and cooling duties, electricity usage, $GWP_{100}$, and FEC are available online[62]. Labeled dark gray lines denote reported impacts for fossil-based production pathways (f1[52]; f2-f4[51]). Labeled light gray lines denote reported impacts for alternative bio-based production pathways (b1[51]; b2[53]; b3[52]). Where $GWP_{100}$ was reported as cradle-to-gate, 1.49 kg $CO_2$-eq./kg was added as end-of-life impacts for consistency with this study and Dunn et al. 2015. Values for all reported MPSPs and impacts before and after adjustment are listed in Supplementary Table 3 and 4. (**C**, **D**) MPSP and (**E**, **F**) $GWP_{100}$ across 2500 fermentation yield-titer combinations at the baseline productivity of the pilot batch scenario (0.66 g/L/h) for neutral (left panel; **C**, **E**) and low-pH (right panel; **D**, **F**) fermentation. Yield is shown as % of the theoretical maximum (%theoretical) scaled to the theoretical maximum yield of 1.31 g/g-glucose-equivalent (based on carbon balance). For a given point on the figure, the *x*-axis value represents the yield, the *y*-axis value represents the titer, and the color and contour lines represent the value of MPSP, $GWP_{100}$. Diamond markers show baseline results for the pilot batch scenario. Source data are provided as a Source Data file.

attempted to recover SA from fermentation broth via direct acidification and crystallization[4,37,40–43]. In these studies, direct crystallization was highlighted as an effective technique to recover SA from fermentation broth. However, acidifying the broth to a lower pH prior to crystallization was required due to fermentation at neutral pH. To date, the maximum SA recovery of 79% was reported via acidification followed by direct crystallization[42]. In the present work, without any additional unit operation and acidification due to low-pH fermentation, direct crystallization could recover SA from the sugarcane juice fermentation broth with an efficiency of 64%. Nevertheless, to achieve high recovery yields and purity of SA and process commercial feasibility, pre-polishing steps in tandem with direct crystallization need to be investigated.

Finally, we performed TEA and LCA to determine the financial viability and environmental benefits of our SA production pipeline. For the laboratory fed-batch scenario, the MPSP, $GWP_{100}$, and FEC of our biorefinery were the lowest reported values thus far for a biorefinery-producing SA. Furthermore, for the pilot batch scenario, the biorefinery's MPSP of $1.37/kg ($1.23–1.54/kg; 5th-95th percentiles, hereafter in brackets) was consistently below the reported market price range of $2.53–2.89/kg[31] (adjusted to 2016$) and near the low end of bio-based SA MPSP values ranging from $1.08–3.63/kg reported in the literature[44–52] (Fig. 4A). Similarly, from LCA, the biorefinery's $GWP_{100}$ of 1.67 kg $CO_2$-eq./kg (1.22–2.17 kg $CO_2$-eq./kg) was consistently below reported values for fossil-based SA production pathways (3.27–12.1 kg $CO_2$-eq./kg[52,53]) by approximately 34–90% across all simulations, and comparable to reported values for alternative bio-based succinic acid production pathways (1.83–2.95 kg $CO_2$-eq./kg[44,53–55]) (Fig. 4B). The biorefinery's FEC of −0.21 MJ/kg (−7.08 to 6.47 MJ/kg) was well below reported values for both fossil-based (59.2–124 MJ/kg[53,54]) as well as alternative bio-based SA production pathways (26–32.7 MJ/kg[53–55]) across all simulations, and below zero in approximately 48% of simulations (Supplementary Fig. 14A). Two studies[45,52] simulating biorefineries producing SA through neutral fermentation reported MPSPs lower than the MPSP range estimated in our work for the pilot batch scenario. For one of these studies, the lower MPSP may be attributed to simulating a lower-cost feedstock (the study assumed a negative cost of acquiring municipal solid waste, with a waste management fee of $35–100/metric ton[45] paid to the biorefinery; the baseline feedstock sugarcane cost assumed in our work was $49.3/dry metric ton, and the baseline values and uncertainty distributions assumed in our work for all parameters are listed in Supplementary Data 1). The second of the two studies had different fermentation performance assumptions (yield of 0.96 g/g, titer of 55.8 g/L, and productivity of 0.77 g/L/h) compared to the fermentation performance achieved at the pilot scale in our work (0.473 g/g, 63.1 g/L, and 0.657 g/L/h, respectively), and applying that study's fermentation assumptions to the biorefinery in our pilot batch scenario would result in a baseline MPSP of $1.17/kg with neutral fermentation and $1.05/kg with low-pH fermentation

(both lower than the MPSP of $1.30/kg[52] reported in that study). A full list of reported MPSP, $GWP_{100}$, and FEC values used for comparison is available in Supplementary Tables 3, 4. These results demonstrate that the currently achieved fermentation performance at the pilot scale enables a bio-based SA production pathway that is more financially viable and far more environmentally beneficial than the fossil-based production pathway and highly competitive with other bio-based pathways.

From the sensitivity analysis we performed for the pilot batch scenario, we found MPSP to be most sensitive to fermentation SA yield and relatively less sensitive to fermentation SA titer. However, we found $GWP_{100}$ and FEC to be more sensitive to fermentation SA titer than to yield, demonstrating the importance of these parameters in achieving a financially viable and environmentally sustainable full-scale process. These findings were consistent with the baseline feed sugarcane cost accounting for ~66% of the biorefinery's annual operating cost (excluding depreciation) and net electricity production resulting in larger offsets to $GWP_{100}$ and FEC (~39% and ~100%, respectively) than to annual operating cost (~19%; Fig. 4A, B and Supplementary Fig. 14A) as higher titer values were associated with reduced separation heating utility demand, allowing the unused steam to be sent to the turbogenerator for electricity production (a detailed description of the simulated co-heat and power generation configuration is available in a previous study[56]).

From the evaluation we performed across the fermentation performance landscape (Fig. 4C–F and Supplementary Fig. 14B, C), we observed low-pH fermentation had consistently lower MPSP, $GWP_{100}$, and FEC values compared to those for neutral fermentation at the same yield, titer, and productivity combinations. This was due to the reduced base and acid requirements associated with low-pH fermentation (Supplementary Data 1). As expected, MPSP benefited from increased yield and titer values for both neutral and low-pH fermentation (Fig. 4C, D). Because of the separation-intensive nature of the biorefinery, the benefits of a higher titer generally outweighed the increased expenses (e.g., higher capital costs for larger equipment due to more diluted streams), but the magnitude of the net benefits diminished with increasing titer. Similarly, at higher yield values, further improvements to yield had diminishing benefits for MPSP. At the baseline fermentation yield-titer combinations for laboratory batch, laboratory fed-batch, and pilot batch scenarios, improvements to yield had a greater benefit for MPSP than comparable relative improvements to titer. For example, at the pilot batch scenario fermentation yield (36.1% of the theoretical maximum) and titer (63.1 g/L), improving yield by 3.6% (a 10% relative increase) would decrease the MPSP by $0.06/kg, while improving titer by 6.3 g/L (a 10% relative increase) would decrease the MPSP by $0.03/kg. However, improvements to titer have much greater potential benefits to $GWP_{100}$ and FEC, as increasing titer would decrease heating and cooling utility demands while increasing yield would increase these environmental impacts

**Table 1 | High succinic acid production by metabolically engineered microorganisms**

| Microorganism | Titer (g/L) | Yield (g/g glucose equivalent) | Productivity (g/L/h) | Medium | pH/neutralizers | Mode | Working volume (L)/Reactor size (L) | Reference |
|---|---|---|---|---|---|---|---|---|
| M. succiniciproducens | 134.25 | 0.82 | 21.3 | CDM with glucose and glycerol | 6.5/ammonia and $Mg(OH)_2$ solutions | Fed-batch | 2.5/6.6 | 6 |
| E. coli | 113.95 | 0.91 | 3.25 | Corn stalk hydrolysate supplemented with yeast extract | 6.6–7.0/$Mg(OH)_2$ and $NH_3 \cdot H_2O$ | Fed-batch | 1.2/3.0 | 66 |
| C. glutamicum | 134 | 1.1 | 2.53 | NaCl solution with glucose and formate | 6.9/KOH | Fed-batch | 0.45/1.4 | 67 |
| Y. lipolytica | 101.4 | 0.37 | 0.70 | YP with glucose | 5.5/unspecified | Fed-batch | Unspecified/1.0 | 11 |
| Y. lipolytica | 110.7 | 0.53 | 0.80 | YP with glycerol | No neutralizer; final pH 3.4 | Fed-batch | 1.0/2.5 | 10 |
| I. orientalis | 109.5 | 0.65 | 0.54 | SC-URA with glucose and glycerol | 3.0/KOH | Fed-batch | 0.1/0.35 | This study |
| I. orientalis | 104.6 | 0.63 | 1.25 | Sugarcane juice | 3.0/KOH | Fed-batch | 0.1/0.35 | This study |
| I. orientalis | 63.1 | 0.5 | 0.66 | Sugarcane juice | 3.0/ammonia solution | Batch | 30/75 | This study |

due to higher electricity consumption: at a fixed titer, an increased succinic acid yield on sugars results in a larger fermentation vessel size required, which necessitates higher mixing power requirements (which increase with fermentation vessel size). In the baseline case for the pilot batch scenario, no natural gas is purchased to satisfy the heating utility demand; natural gas is purchased solely for the gas-fired dryer that removes moisture from crystallized succinic acid. In fact, enough steam is produced to completely satisfy the heating and power utility demands and produce excess electricity (using a turbogenerator) to be sold back to the grid and displace the $GWP_{100}$ and FEC impacts associated with grid electricity (Fig. 4E, F and Supplementary Fig. 14B, C; a detailed description of the simulated co-heat and power generation configuration is available in a previous study[56]). Collectively, this evaluation of MPSP, $GWP_{100}$, and FEC across the potential fermentation performance landscape reveals that while SA yield is more critical to the economics of the biorefinery than titer at the current state of the technology, continued improvements to titer represent the greatest opportunity to reduce environmental impacts.

In conclusion, by employing several metabolic engineering strategies, we have obtained an *I. orientalis* strain that can produce more than 100 g/L of SA (at the maximum solubility of SA) using minimal media at low pH (pH 3). This is also the best overall performance to date for a yeast strain that used the carbon-fixing rTCA pathway. Furthermore, our TEA and LCA performed under uncertainty demonstrate that the currently achieved fermentation performance at the pilot scale enables an end-to-end bio-based SA production pathway that is more financially viable and far more environmentally beneficial than the fossil-based production pathway and highly competitive with other bio-based pathways. Further improvements to the sustainability of this pathway may be achieved through a higher fermentation yield and titer. In particular, we are attempting to couple the rTCA pathway with the glyoxylate shunt pathway, which theoretically enables the maximum yield of 1.71 mol/mol glucose[57]. Furthermore, supplementation of crude glycerol, which is the main byproduct produced during the transesterification process in biodiesel plants[20], to the sugarcane juice medium can potentially increase titer and yield, as demonstrated with the fermentations using minimal medium with pure glucose and pure glycerol. Overall, our study presents an end-to-end pipeline for the economical production of SA from sugars at low pH and illustrates how agile and robust system analyses could enable bioprocess development for sustainable production of organic acids. We anticipate this pipeline is potentially applicable to the production of other organic acids at low pH, such as muconic acid or 3-hydroxypropionic acid.

## Methods

### Strains, media, and materials

All strains used in this study are described in Supplementary Table 5. *E. coli* DH5α was used to maintain and amplify plasmids and was grown in Luria Bertani medium (1% tryptone, 0.5% yeast extract, 1% NaCl) at 37 °C with ampicillin (100 μg/mL). *I. orientalis* SD108 and *S. cerevisiae* HZ848 were propagated at 30 °C in YPAD medium consisting of 1% yeast extract, 2% peptone, 0.01% adenine hemisulphate, and 2% glucose. Recombinant *I. orientalis* strains were cultured in Synthetic Complete (SC) dropout medium lacking uracil (SC-URA). Sugarcane juice medium for fermentation was prepared by diluting the sugarcane juice by 2-fold and dissolving ammonium sulfate and magnesium sulfate at concentrations of 5 g/L and 1 g/L, respectively. LB broth, bacteriological grade agar, yeast extract, peptone, yeast nitrogen base (w/o amino acid and ammonium sulfate), and ammonium sulfate were purchased from Difco (BD, Sparks, MD), while complete synthetic medium was obtained from MP Biomedicals (Solon, OH). All restriction endonucleases and Q5 DNA polymerase were purchased from New England Biolabs (Ipswich, MA). QIAprep Spin Miniprep Kit was purchased from Qiagen (Valencia, CA), and Zymoclean Gel DNA Recovery

Kit and Zymoprep Yeast Plasmid Miniprep Kits were purchased from Zymo Research (Irvine, CA). All other chemicals and consumables were purchased from Sigma (St. Louis, MO), VWR (Radnor, PA), and Fisher Scientific (Pittsburgh, PA). Oligonucleotides including gBlocks and primers were synthesized by Integrated DNA Technologies (IDT, Coralville, IA).

## Plasmids and strains construction

The plasmids, gBlocks, codon-optimized genes, and primers are listed in Supplementary Tables 6, 7, and 8 and Supplementary Data 2, respectively. Genes were codon-optimized and synthesized by Twist Bioscience (San Francisco, CA). Plasmids were generated by the DNA assembler method in *S. cerevisiae*[58], and Gibson assembly[59] and Golden Gate assembly[60] in *E. coli*. For DNA assembly, 100 ng of PCR-amplified fragments and restriction enzyme digested backbone were co-transformed into *S. cerevisiae* HZ848 via the electroporation method. Transformants were plated on SC-URA plates and incubated at 30 °C for 48–72 h. Yeast plasmids were isolated and transformed to *E. coli* for enrichment. *E. coli* plasmids were extracted and verified by restriction digestion. The details of plasmid construction procedures are described in Supplementary Method 1, while the details of strain construction procedures are described in Supplementary Method 2. The lithium acetate-mediated method was used to transform yeast strains with plasmids and donor DNA fragments[61].

## Fermentation experiments

For shake flask fermentations, single colonies of *I. orientalis* strains were inoculated into 2 mL of liquid YPAD medium with 20 g/L of glucose and cultured at 30 °C for 1 day. Then, the cells were subcultured in 2 mL of liquid SC-URA medium with 20 g/L of glucose and grown at 30 °C for 1 day to synchronize the cell growths. Cells were then transferred into 20 mL of SC-URA liquid medium with 50 g/L glucose, 50 g/L glucose and 20 g/L glycerol, or 70 g/L glucose in 125 mL Erlenmeyer flask. Cells were diluted to an initial $OD_{600}$ of 0.2, and 10 g/L of calcium carbonate were supplemented in the fermentations. The cells were cultivated at 30 °C at $0.14g$ (oxygen limited condition) or $0.87g$ (aerobic condition). Samples were collected every 24 h for HPLC analysis. Shake flask fermentations were conducted with three biological replicates.

For fed-batch fermentations in bench-top bioreactors (DASbox, Eppendorf, Hamburg, Germany), single colonies of *I. orientalis* strains were inoculated into 2 mL liquid YPAD medium with 20 g/L of glucose and cultured for 1 day. Then, the cells were subcultured into 2 mL liquid SC-URA medium with 20 g/L of glucose and grown for 1 day. 1 mL of cells was then added into 100 mL of liquid SC-URA medium with 50 g/L glucose and 20 g/L glycerol or 100 mL of sugarcane juice medium in DASbox. The cells were cultivated at 30 °C with $10g$. pH was maintained at 3 using 4 N HCl and 4 N KOH. Industrial-grade $CO_2$ and $O_2$ gasses were continuously sparged into the bioreactors at flow rates of 0.333–0.667 vvm and 0.167 vvm, respectively. One drop of Antifoam 204 was added to control foaming if necessary. For fermentations using pure glucose and glycerol, after the initial glucose and glycerol were depleted, additional glucose and glycerol were added to the bioreactors. For fermentation using sugarcane juice, after the initial sugars were depleted, sugarcane juice, which was concentrated by boiling, was added to the bioreactors. Samples were collected every 24 h for HPLC analysis. Fed-batch fermentations were conducted with two biological replicates.

## Analytic methods

Extracellular glucose, glycerol, pyruvate, succinate, and ethanol concentrations of fermentation broths were analyzed using the Agilent 1200 HPLC system equipped with a refractive index detector (Agilent Technologies, Wilmington, DE, USA) and Rezex ROA-Organic Acid H+ (8%) column (Phenomenex, Torrance, CA, USA). The column and

detector were run at 50 °C, and 0.005 N $H_2SO_4$ was used as the mobile phase at a flow rate of 0.6 mL/min.

## qPCR analysis

*I. orientalis* cells were inoculated in YPAD medium and grown at 30 °C with constant shaking at $0.87g$ overnight. The cells were then subcultured into fresh YP medium with 20 g/L glycerol with the initial $OD_{600}$ of 0.2 and grown until the $OD_{600}$ reached 1. Cells were collected from 1 mL of culture, and total RNA was extracted using the RNeasy mini kit from Qiagen (Valencia, CA). DNase treatment of RNA was performed using the RNase-Free DNase Set from Qiagen. cDNA synthesis was performed using the iScript™ Reverse Transcription Supermix from Biorad, and iTaq Universal SYBR Green Supermix from Biorad was used for qPCR following the manufacturer's protocol. Primers for qPCR were designed using the IDT online tool (Primer Quest). The endogenous gene *alg9*, encoding a mannosyltransferase, was used as the internal control. Expression of the selected gene was normalized by the *alg9* expression level. Raw data was analyzed using QuantStudio™ Real-time PCR software from Applied Biosystems. qPCR analysis was performed with two biological duplicates.

## Metabolic flux analysis

To determine the fluxes of glucose consumption, pyruvate production, and succinate production, strain SA/MAE1/pdcΔ/gpdΔ was grown overnight in yeast nitrogen base (YNB) medium without amino acids consisting of 5% glucose and then inoculated in YNB medium at $OD_{600}$ of 0.1. At 0, 24, and 53 h, $OD_{600}$ was measured, and the supernatant was collected. The supernatant was then diluted 50-fold in a solution of 40:40:20 methanol:acetonitrile:water. Glucose, pyruvate, and succinate were quantified with external calibration standard by LC-MS[62]. LC analysis was performed using a Vanquish UHPLC system (Thermo Fisher) and Xbridge BEH Amide HILIC column (Waters) with a gradient of 25 min from acetonitrile to pH 9.5 aqueous buffer. The injected sample volume was 5 μL. LC was coupled by electrospray ionization (± 3.3 kV) to a Q-Exactive Plus mass spectrometer (Thermo Fisher). Raw LC/MS data were converted to mzXML format by msconvert from ProteoWizard. Peak extraction of the raw data for cells growing in unlabeled media was performed using the ElMaven software package (https://elucidatainc.github.io/ElMaven). A conversion factor of 0.6 grams dry weight per $OD_{600}$ per liter was used to convert $OD_{600}$ to a cell dry weight unit.

For $^{13}C$ isotope tracing analysis, yeast was cultured in media with U-$^{13}C_6$ glucose or 1,2-$^{13}C_2$ glucose (Cambridge Isotope Laboratories, Tewksbury, MA, USA) at 50% enrichment. Strain SA/MAE1/pdcΔ/gpdΔ was first grown overnight and then inoculated into fresh media at $OD_{600}$ of 1. The cultures were allowed to grow for about 36 h to reach $OD_{600}$ of 6. For intracellular metabolite extraction, about 900 μL of cell culture was quickly vacuum filtered through a GVS Magna™ Nylon membrane filter with 0.5 μm pore size (Fisher Scientific, Pittsburgh, PA), quenched in 1 mL of ice-cold solution of 40:40:20 methanol:acetonitrile:water with 0.5% formic acid for about 2 min, and then neutralized with 88 μL of ammonium bicarbonate. The extracts were centrifuged at 17000 x $g$, and the supernatants were analyzed by LC-MS. For amino acid analysis, the pellets from metabolite extractions were washed with water and hydrolyzed in 100 μL 2 M HCl at 80 °C for 1 h. Then, 10 μL of the hydrolysate supernatant was dried under pure nitrogen and redissolved in 100 μL of solution of 40:40:20 methanol:acetonitrile:water and analyzed by LC-MS. For LC-MS data analysis, the data was converted to mzXML by msconvert from ProteoWizard, and mass peaks were then picked by the ElMaven software package (https://elucidatainc.github.io/ElMaven). $^{13}C$ natural isotope abundance was corrected using the accucor R package (https://github.com/lparsons/accucor).

$^{13}C$ metabolic flux analysis was done with a customized core atom mapping model with redox balance (Supplementary Data 3) in the INCA1.9 Suite[63]. Flux solution that best fits the mass isotope

distribution of 26 metabolites was obtained under the constraint of glucose, pyruvate, and succinate fluxes and growth rate. Flux lower and upper bounds were obtained using parameter continuation.

## Pilot-scale fermentation

SA production using sugarcane juice was scaled up from DASbox (maximum working volume of 250 mL and reactor volume of 350 mL) to pilot-scale fermenter (maximum working volume of 60 L and reactor volume of 75 L). A single colony of strain g3473Δ/PaGDH-DAK/ScSUC2 was inoculated into 2 mL of liquid YPAD medium with 20 g/L of glucose and cultured for 1 day. Then, the cells were subcultured into 1 L of liquid YPAD medium with 20 g/L of glucose and grown for 1 day. 1 L of culture was then added into the pilot-scale fermenter containing 30 L of sugarcane juice medium. The cells were cultivated at 30 °C with agitation and air flow rate varied to maintain 8% DO. pH was maintained at 3 using 28% ammonium hydroxide solution. Industrial-grade $CO_2$ was continuously sparged into the bioreactors at flow rates of 0.2 vvm. Antifoam 204 was added to control foaming if necessary. The rheological properties of the fluid except for 2× concentrated sugarcane juice were assumed to be the same as those of water. The scale-up criteria were considered to maintain similar power input per unit volume ($P.V^{-1}$, or the mean specific energy dissipation rate) and the same Reynolds number. For the DASbox, Reynolds number and P/V were calculated based on the density of sugarcane juice, impeller tip velocity, and impeller diameter. For the pilot-scale fermenter, the agitation was fixed to maintain $P.V^{-1}$ and Reynolds number values similar to those for the DASbox. The calculations related to tip speed, Reynolds number, and power consumption were performed via standard formulas (as illustrated in Supplementary Method 3) for the DASbox and for the pilot-scale fermenter. The DO% set-point was maintained at 8% saturation via cascading it with air flow rate into the pilot-scale fermenter. The geometrical specifications of the DASbox and the pilot-scale fermenter are described in Supplementary Table 9.

## SA crystallization, recovery, and purity

The parameters that could influence the crystallization process were identified as seed loading, seed loading temperature, time, and agitation. The crystallization temperature was fixed at 0 °C. These parameters were optimized utilizing synthetic solutions comprising SA concentrations of 100 g/L, 200 g/L, and 250 g/L as described in Supplementary Method 4. These optimized values were estimated as 1% w/v, 10 °C, 4 h, and 0.56$g$ for seed loading, seed loading temperature, time, and agitation, respectively (Supplementary Table 10). They were applied to directly crystallize SA from sugarcane juice fermentation broth at pH 3 sequentially. The sugarcane juice fermentation broth contained yeast cells, insoluble and soluble macromolecules, and other metabolites (not characterized). For SA crystallization, 100 mL of the batch fermentation broth in the pilot-scale fermenter with an SA initial concentration of 63 g/L was heated to 80 °C and retained for 30 min on a hotplate stirrer (Thermo Fischer Scientific). The cells and suspended solids were removed from the heated broth via filtration (Whatman Grade GF/A, binder-free, Glass Microfiber Filters (Cytiva)). Afterward, to concentrate the broth to 50% of its original volume, the obtained broth was subjected to vacuum distillation (Rotavapor, BUCHI UK Ltd) at 60 °C. The resulting broth was subjected to direct crystallization stage 1 at 0 °C with an agitation of 0.56$g$ in a temperature-controlled shaker for 4 h. The crystallized SA was recovered from the broth via filtration using Glass Microfiber Filters (Whatman Grade GF/A, Cytiva). The filtrate was then concentrated to 50% of its initial volume of crystallization 1 via vacuum distillation (Rotavapor, BUCHI UK Ltd) at 60 °C. The obtained concentrate was subjected to direct crystallization stage 2 under the same optimized conditions of crystallization stage 1. The SA crystals formed were collected after 4 h by filtration via Glass Microfiber Filters (Whatman Grade GF/A, Cytiva) followed by drying of crystals at 70 °C for 1 day. The SA recovery overall and stage-wise was calculated via Eq. 1

given below.

$$\% \text{ recovery} = \frac{\text{Weight of SA recovered in stages/overall}}{\text{Initial SA present in solution}} \times 100 \quad (1)$$

The SA crystals recovered via crystallization in stage 1 and stage 2 were separated from the fermentation broth by simple filtration. From both crystallization stages, the recovered SA crystals were washed with cold Millipore water (4 °C) to eliminate any impurities from the crystal surface. Later, the washed SA crystals were dried at 50 °C for 24 h for subsequent analysis. To measure the SA purity, a solution of dried crystals (stages 1 and 2) of concentration 5 g/L (0.1 g in 20 mL) was made utilizing Millipore water. Later, the SA weight in the crystal was calculated by comparing the peak area against the analytical grade SA solution of the same concentration procured from Sigma-Aldrich, USA. The analysis was carried out in duplicate and the purity was represented by Eq. 2.

$$\% \text{ purity} = \frac{\text{Weight of SA in crystal recovered}}{\text{Weight of crystal taken}} \times 100 \quad (2)$$

## Techno-economic analysis and life cycle assessment

To perform the techno-economic analyses and life cycle assessment presented in this work, we leveraged BioSTEAM, an open-source platform in Python[64,65]. Briefly, influent and effluent streams of each unit are simulated by BioSTEAM and coupled with operating parameters and equipment cost algorithms for unit design and cost calculations. The MPSP was estimated in 2016$. The functional unit of the TEA and LCA was 1 kg of succinic acid in the product stream. The scope of the LCA was cradle-to-grave for $GWP_{100}$ (assuming the end-of-life degradation of the product succinic acid entirely to $CO_2$) and cradle-to-gate for FEC. The system boundary of the LCA is depicted in Supplementary Fig. 17. Uncertainty distributions of key parameters are available in Supplementary Data 1. All assumed environmental impacts and prices (with references), Python scripts for BioSTEAM and the biorefinery (including biorefinery setup and system analyses) as well as a system report (including the detailed process flow diagram, stream composition and cost tables, unit design specifications, and utilities for the baseline simulations) are available in the online repository[64].

### Reporting summary

Further information on research design is available in the Nature Portfolio Reporting Summary linked to this article.

## Data availability

The data supporting the findings of this work are available within the paper and its Supplementary Information files. The DNA sequences of all plasmids used in this study are provided as Supplementary Data 4. Source data are provided with this paper.

## Code availability

Python scripts for BioSTEAM and the biorefinery as well as a system report can be found at Github [https://github.com/BioSTEAM DevelopmentGroup/Bioindustrial-Park/tree/master/biorefineries/ succinic].

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

## Acknowledgements

This material is based on research sponsored by the U.S. Department of Energy award DE-SC0018420 and the Air Force under agreement number FA8650-21-2-5028. The U.S. Government is authorized to reproduce and distribute reprints for Governmental purposes notwithstanding any copyright notation thereon. The views and conclusions contained herein are those of the authors and should not be interpreted as necessarily representing the official policies or endorsement, either expressed or implied, of the Air Force or the U.S. Government. This publication was made possible with the support of The Bioindustrial Manufacturing and Design Ecosystem (BioMADE); the content expressed herein is that of the authors and does not necessarily reflect the views of BioMADE. We thank the Integrated Bioprocessing Research Laboratory for assisting with scale-up and piloting. Thanks to Kristen Eilts for conducting the HPLC analyses for the pilot plant runs. The online tool BioRender (https://www.biorender.com/) was used to create Fig. 1.

## Author contributions

V.G.T. and H.Z. conceived and designed the study. V.G.T. constructed and characterized all *I. orientalis* strains. S.S., S.M. and V.G.T. performed scale-up and piloting. S.M. performed succinic acid recovery. S.S.B. performed biorefinery design, modeling, techno-economic analysis, and life cycle assessment. Y.S. performed metabolic flux analysis and Y.S. and J.D.R. analyzed the data. J.L.A. performed literature survey on reported techno-economic analyses for bio-based succinic acid. B.A.C., S.-I.T. and Z.F. assisted with constructions of plasmids and strains. V.G.T., S.M., S.S.B., V.S., J.S.G. and H.Z. wrote the manuscript with input from all other authors.

## Competing interests

A provisional patent application on the metabolic engineering of *I. orientalis* for succinic acid production (US63/354, 994, V.G.T and H.Z.) has been filed based on this study. Other authors claim no competing interests.
