## [Peer Review File · Nature Communications]

An End-to-end Pipeline for Succinic Acid Production at an Industrially Relevant Scale using *Issatchenkia orientalis*Reviewers' Comments:

Reviewer #1:

Remarks to the Author:

In this manuscript, the metabolic engineering of *Issatchenkia orientalis*, a non-conventional yeast with good tolerance to acidic conditions, was systematically studied for the production of succinic acid. The recombinant strains produced 109.5 g/L succinic acid in minimal medium and 104.6 g/L in sugarcane juice medium at pH 3 in fed-batch fermentations. A downstream processing comprising of two-stage distillation and crystallization was developed for the direct recovery of succinic acid from fermentation broth without acidification. The techno-economic analysis and life cycle assessment of the low-pH succinic acid production process indicate that life cycle greenhouse gas emissions could be reduced by 34-90% relative to fossil-based production processes.

It is an interesting research and worth publication.

To my knowledge, this is the first publication over 100 g/L succinic acid in minimal medium at pH as low as 3, and with acidic tolerance microbes. This result is interesting for both academic and industry.

1. In figure 2, the initial concentrations of glucose and glycerol were around 40 g/L and 20 g/L separately. The concentrations of glucose and glycerol after each fed were around 40 g/L and 20 g/L separately. Why and how were the concentrations 40 g/L and 20 g/L chosen for glucose and glycerol separately?

2. "The codon optimized PaGDH and endogenous DAK were overexpressed". Were PaGDH and DAK on the genome or a plasmid? How were the expression of PaGDH and DAK controlled in the recombinant?

3. Line 201-202, "batch fermentation in bench-top bioreactor with size of 0.3 L and working volume of 0.1 L"

Line 238-239, "batch fermentation was performed in a pilot-scale bioreactor with size of 75 L and working volume of 30 L"

The ratio of working volume to size were very low. Why was the ratio was so low? Normally, the concentration of glucose in the feed supplement was very high in industrial fed-batch fermentation. The ratio of working volume to size was normally no less that 70% in industrial fed-batch fermentation.

4. Line 202, "continuous sparging of O₂ and CO₂". Was the concentration of CO₂ in the fermentation broth measured and controlled? How much was the concentration of CO₂ in the fermentation broth? Were there optimum concentrations of O₂ and CO₂ for the production of succinic acid?

5. Line 251-252 "Thus, via two-stage vacuum distillation and crystallization, the overall SA recovery yield of 64.0% from low-pH fermentation broth was obtained." 64% is not high. In order to increase the recovery rate, other operation units should be added into the downstream processing. So, it might not be adequate to calculate the life cycle greenhouse gas emission with such process.

Reviewer #2:

Remarks to the Author:

The study developed *I. orientalis* with superior tolerance to highly acidic conditions, through deletion of byproduct pathways, transport engineering, and expanding the substrate scope. The resulting strain could produce succinic acid at high titers at low pH levels using sugar-based media in fed-batch fermentation and pilot-scale fermentation. A two-stage downstream processing technique was developed, and techno-economic analysis and life cycle assessment indicate the cost-effective and environmentally friendly potential of the process. Although the research is meaningful, I think the paper needs to resolve quite a few issues as listed below:

Comments:

1. Bacteria has higher titers, yields, and productivities than the engineered yeasts (Page 3 Paragraph 2 and Table 1), then why don't you try to engineer bacteria to be more acid-resistant strain? Because it seems that engineering modifications to bacteria can achieve higher SA productivity while reducing

downstream purification steps, lowering cost and environmental impact.

2. There should be more information on the metabolic pathways of SA production in *I. orientalis* in the introduction part, especially those pathways related to the modifications made in the study, including transporters and substrates.

3. How to define the oxygen-limited condition? Why the flask-scale fermentation were conducted under oxygen-limited conditions but aerobic condition? From Fig. S4, aerobic condition also increase the glucose consumption rate, why didn't you use aerobic condition when using glucose as single carbon source?

4. The symbols and abbreviations used in the figures should be explained at their first mention in the figure. For example, the abbreviations in Fig.1 (SUC2, HXK, etc.) should be explained.

5. From Fig. S7, different initial glucose concentration lead to different SA yield. How do you decide glucose concentration of 50 g/L? Did you try other concentration? Similarly, how do you decide the substrate ratio of 50 g/L glucose and 20 g/L glycerol? Did you try other ratio?

6. The functional unit and system boundary of LCA should be defined.

7. Did you analyse the purity of your SA crystal, and did you consider purity when doing TEA? As the purity will influence the economic analysis in TEA.

Reviewer #3:

Remarks to the Author:

This manuscript studied metabolic engineering strategies, performed both fermentation experiments and TEA and LCA on biological conversion of sugar to succinic acid (SA) production pathway. The study is inclusive studies of both fermentation experiments and cost/sustainability analysis to guide R&D. In that regard, the manuscript is a comprehensive study of SA conversion pathways. My general comments are:

- 1) Several previous literatures have already demonstrated high yield and high titer, what is uniqueness and values of metabolic engineering practices here?
- 2) I understand glycerol and sucrose can improve overall yield and titer, but those two substrates were not discussed extensively for the readers to fully interpretate the supply chain constraints and impacts, my suspension is whether those substrates are only for high yield demonstration? If that is the case, is it practical?
- 3) I do have some reservation on the uncertainty analysis. For instance, typical Monte Carlo analysis requires over 5,000 runs, so I am not sure whether 2,000 runs performed in this study would be adequate;
- 4) the main text called out tables and figures from supplementary information more frequently than other papers, the readers have to cross check the two documents almost all the time. What is the point of adding that valuable information into supplementary information then? Would it be more convenient just to combine into the main text?

Specific comments are also listed below:

- Title: "pipeline" is an interesting choice of word.
- Table 1: other than larger reactor demonstration, what are the unique contribution from this study on improving SA fermentation technologies? The fermentation performance from reference 65 seems more favorable. Also why 75L reactor resulted in a significant yield reduction?
- Page 9, "we designed and simulated biorefineries across the fermentation performance landscape (i.e., 2,500 yield-titer combinations each across a range of productivities for both neutral and low-pH fermentation) to set and prioritize targets for further improvements to financial viability and environmental sustainability". The definition of variable distribution should be based on experimental

data or warranted experiences, not with a design. This will directly impact the credibility of the cost distribution.

- Page 10, "We engineered the strains by deletion of byproduct pathways, transport engineering to enable SA export and limit SA import, and expansion of the substrate scope to allow utilization of glycerol and sucrose." Supply constraints and price impacts are not discussed in the TEA or LCA, which should be critical to address the questions on why including these two substrates to SA fermentation/production.
- Page 12, "; the baseline feedstock sugarcane cost assumed in our work was \$49.3/dry metric ton)." What is the basis/credible reference to assume this feedstock cost? The cost number seems low to me.
- Page 12, "The second of the two studies had different fermentation performance assumptions (yield of 0.96 g/g, titer of 55.8 g/L, and productivity of 0.77 g/L/h) compared to the fermentation performance achieved at the pilot scale in our work (0.473 g/g, 63.1 g/L, and 0.657 g/L/h, respectively)." Surprised to see such a low yield. Any explanation on the yield differences between this study and the study with 0.96 g/g yield.
- Page 13, "...while SA yield is more critical to the economics of the biorefinery than titer at the current state of the technology, continued improvements to titer represent the greatest opportunity to reduce environmental impacts." I enjoyed this discussion, but this is a bit qualitative. IF you could add quantitative discussion on at what specific level of yield the yield becoming critical to cost and GHG, that would be helpful to the readers.
- Page 14, "Crude glycerol, as a waste, is also a no-cost or low-cost substrate, ..." I totally disagree with this assumption. Nothing is free.

We thank all the reviewers for their thoughtful comments. We addressed their questions and concerns point-by-point as described below. We also followed the journal format requirement and editorial policy. All the corresponding changes are highlighted in red in the revised manuscript.

Reviewer #1:

In this manuscript, the metabolic engineering of *Issatchenkia orientalis*, a non-conventional yeast with good tolerance to acidic conditions, was systematically studied for the production of succinic acid (SA). The recombinant strains produced 109.5 g/L succinic acid in minimal medium and 104.6 g/L in sugarcane juice medium at pH 3 in fed-batch fermentations. A downstream processing comprising of two-stage distillation and crystallization was developed for the direct recovery of succinic acid from fermentation broth without acidification. The techno-economic analysis and life cycle assessment of the low-pH succinic acid production process indicate that life cycle greenhouse gas emissions could be reduced by 34-90% relative to fossil-based production processes.

It is an interesting research and worth publication.

To my knowledge, this is the first publication over 100 g/L succinic acid in minimal medium at pH as low as 3, and with acidic tolerance microbes. This result is interesting for both academic and industry.

1. In figure 2, the initial concentrations of glucose and glycerol were around 40 g/L and 20 g/L separately. The concentrations of glucose and glycerol after each fed were around 40 g/L and 20 g/L separately. Why and how were the concentrations 40 g/L and 20 g/L chosen for glucose and glycerol separately?

We have added the following results to the revised manuscript to describe how we chose to use 50 g/L of glucose with 20 g/L of glycerol.

Lines 194-200: “We also compared the SA production in strain g3473Δ/PaGDH-DAK using 50 g/L of glucose and different glycerol concentrations of 10 g/L, 20 g/L, and 30 g/L (**Fig. S8**). As expected, SA titer increased as the initial amount of glycerol increased; thus, a comparison between titers was not meaningful. Also, the yield was lowest when 10 g/L of glycerol were used. No significant difference in yields was observed when 20 g/L and 30 g/L of glycerol were used; however, the productivity was highest when 20 g/L of glycerol were used. Thus, we considered 20 g/L of glycerol to be the optimal concentration to use with 50 g/L of glucose for the SA production.”

2. “The codon optimized PaGDH and endogenous DAK were overexpressed”. Were PaGDH and DAK on the genome or a plasmid? How were the expressions of PaGDH and DAK controlled in the recombinant?

The PaGDH and DAK were integrated into the genome and expressed using strong constitutive promoters.

3. Line 201-202, “batch fermentation in bench-top bioreactor with size of 0.3 L and working volume of 0.1 L.” Line 238-239, “batch fermentation was performed in a pilot-scale bioreactor with size of 75 L and working volume of 30 L.” The ratio of working volume to size was very low. Why was the ratio so low? Normally, the concentration of glucose in the feed supplement was very high in industrial fed-batch fermentation. The ratio of working volume to size was normally no less than 70% in industrial fed-batch fermentation.

For the bench-top bioreactors (DASbox, Eppendorf, Hamburg, Germany), we followed the user manual and used the working volume of 0.1 L. Furthermore, the scale-up studies were carried out utilizing BioFlo 54-002 Fermentor/Bioreactor (75 L total volume) from NEW BRUNSWICK SCIENTIFIC CO., INC., USA. As per the supplier manual, for this bioreactor, the minimum and maximum working volumes of the total reactor volume were 25% and 75%, respectively. Thus, the 75-L bioreactor was flexible to work in the working volume range of 18.75 L to 56.26 L. In addition, the substrate used for this study (sugarcane juice) was obtained from sugarcane grown in Florida, which was shipped to the University of Illinois. We had limited substrate for all the shake flasks, bench-top bioreactors, and pilot-scale studies. Furthermore, the present scale-up work was the demonstration at the pre-commercialization stage and a key enabler in commercialization. Before commercialization, several additional testings need to be performed to enable a steady and smooth transition into the market.

4. Line 202, “continuous sparging of O₂ and CO₂.” Was the concentration of CO₂ in the fermentation broth measured and controlled? How much was the concentration of CO₂ in the fermentation broth? Were there optimum concentrations of O₂ and CO₂ for the production of succinic acid?

Unfortunately, due to the lack of a head-space CO₂ analyzer and a CO₂ probe, the CO₂ concentrations in the fermentation broth or the gas exhaust were not measured. For the bench-top reactors, we tested two different sets of O₂ and CO₂ flow rates (0.167 vvm O₂ and 0.333 vvm CO₂, and at 0.167 vvm O₂ and 0.667 vvm CO₂). We found that higher CO₂ flow rate led to higher titers in batch fermentations (**Fig. S10**), and at 0.167 vvm O₂ and 0.667 vvm CO₂ we were able to achieve more than 100 g/L of SA in fed-batch fermentations (**Fig. 3**). From these fed-batch fermentations in the bench-top reactors, we found the DO varied between 5-8%; thus, the optimum concentrations of DO < 8% were used for pilot-scale fermentation. At the 75-L scale, CO₂ was continuously sparged at only 0.2 vvm due to equipment limitation, while the dissolved oxygen (DO) was maintained at 8% saturation by cascades. We will implement CO₂ balance to determine the optimal CO₂ concentrations in our future studies.

5. Line 251-252 “Thus, via two-stage vacuum distillation and crystallization, the overall SA recovery yield of 64.0% from low-pH fermentation broth was obtained.” 64% is not high. In order

to increase the recovery rate, other operation units should be added into the downstream processing. So, it might not be adequate to calculate the life cycle greenhouse gas emission with such process.

The SA crystal purity and yield experimentally recovered from fermentation broth were >80% and 64%, respectively. This was due to higher solubility of SA in multicomponent fermentation broth. The higher yield of SA was obtained at a higher SA titer value (synthetic solution result given in SI). Furthermore, the loss of SA from the crystal surface during washing could not be ignored. The spectrum of the work was not to cover the detailed scale-up of the crystallization operation. In fact, the key finding was that the crystallization of untreated broth was successful (because of low pH production of succinic acid by *I. orientalis*) without further acidification or generating any secondary molecule or pollutant. Indeed, an apparent amount of SA remained miscible in the fermentation broth, and more investigation and optimization are required to obtain higher recovery yield and purity. Additionally, the presence of other solutes, and coloring impurities in the fermentation broth has been a major bottleneck to the crystallization process and can be attributed to the lower SA recovery yield during crystallization.

Furthermore, we had already included more downstream unit operations in our simulated biorefinery than in the experimental setup. Just as the reviewer recommended, we had a third vacuum distillation-crystallization stage as opposed to just two vacuum distillation-crystallization stages in the experimental setup, as shown in **Fig. S13**. We used the experimental data from the crystallization experiments to determine how much SA each crystallizer could recover. We conservatively evaporated up to 250 g/L of SA before each crystallization step to justify our use of the experimental data. While the crystallization efficiency of 64% was achieved experimentally using two vacuum distillation-crystallization stages, the overall SA recovery in the simulated biorefinery's separation process was 98.33% of the SA in the fermentation broth.

Reviewer #2:

The study developed *I. orientalis* with superior tolerance to highly acidic conditions, through deletion of byproduct pathways, transport engineering, and expanding the substrate scope. The resulting strain could produce succinic acid at high titers at low pH levels using sugar-based media in fed-batch fermentation and pilot-scale fermentation. A two-stage downstream processing technique was developed, and techno-economic analysis and life cycle assessment indicate the cost-effective and environmentally friendly potential of the process. Although the research is meaningful, I think the paper needs to resolve quite a few issues as listed below:

Comments:

1. Bacteria have higher titers, yields, and productivities than the engineered yeasts (Page 3 Paragraph 2 and Table 1), then why don't you try to engineer bacteria to be more acid-resistant strain? Because it seems that engineering modifications to bacteria can achieve higher SA productivity while reducing downstream purification steps, lowering cost and environmental impact.

Indeed, it is possible to engineer bacteria to be more acid-resistant using approaches such as adaptive laboratory evolution (ALE). For example, previous works showed that ALE could improve the growth rates and fitness of *Escherichia coli* under pH 4.6 and pH 5.5^{1,2}. The media used for ALE were adjusted to low pH using strong acids like HCl and H₂SO₄. However, to our knowledge, no bacteria has been successfully engineered to tolerate pH 3. Moreover, tolerance to high SA concentration, in addition to low pH, is equally important. At high SA concentration at low pH, SA will be in undissociated form and can cross the cell membrane by passive diffusion. Once in the cytoplasm at neutral pH, SA will dissociate into the anion form (succinate) and protons. Accumulation of succinate can disrupt the cellular osmolarity, lowering cell growth and viability³. To our knowledge, engineering bacteria to tolerate more than 100 g/L of SA at pH as low as pH 3 has not been achieved. Furthermore, if such acid-tolerant bacterial strains can be engineered, these strains might suffer a decrease in fermentative performance. Finally, use of bacteria may encounter a phage contamination issue.

References:

1. Harden, M. M. et al. Acid-adapted strains of *Escherichia coli* K-12 obtained by experimental evolution. *Appl. Environ. Microbiol.* **81(6)**, 1932-1941 (2015).
2. Du, B. et al. Adaptive laboratory evolution of *Escherichia coli* under acid stress. *Microbiol.* **166(2)**, 141 (2020).
3. Warnecke, T. & Gill, R.T. Organic acid toxicity, tolerance, and production in *Escherichia coli* biorefining applications. *Microb. cell factories* **4**, 25 (2005).

2. There should be more information on the metabolic pathways of SA production in *I. orientalis* in the introduction part, especially those pathways related to the modifications made in the study, including transporters and substrates.

We have added additional information to the Introduction part in the revised manuscript.

Lines 71-73: “We previously engineered *I. orientalis* SD108 to produce SA using the reductive TCA pathway (rTCA) at a titer of 11.6 g/L in batch cultures using shake flasks¹³. The pathway consisted of four enzymes: pyruvate carboxylase (PYC), malate dehydrogenase (MDH), fumarase (FUMR), and heterologous fumarate reductase (FRD) (**Fig. 1B**).”

Lines 76-77: “by deletion of byproduct pathways, transport engineering, and expanding the substrate scope.”

3. How to define the oxygen-limited condition? Why the flask-scale fermentation were conducted under oxygen-limited conditions but aerobic condition? From Fig. S4, aerobic condition also increase the glucose consumption rate, why didn't you use aerobic condition when using glucose as single carbon source?

For fermentations using shake flasks, aeration is defined by agitation as indicated in the “Fermentation experiments” in the “Materials and methods” section. In particular, for oxygen-limited conditions, low RPM, such as 100 RPM, is typically used; on the other hand, for aerobic conditions, higher RPM, such as 250 RPM, is used.

We have included the fermentation of strain SA/MAE1/pdcΔ/gpdΔ that used glucose as a single carbon source under aerobic conditions.

Lines 141-146: “We also tested fermentation of strain SA/MAE1/pdcΔ/gpdΔ using SC-URA medium with 50 g/L glucose under aerobic conditions. Interestingly, while the rTCA pathway is a fermentative pathway and higher aeration might channel more carbon flux into the TCA cycle for aerobic respiration, we observed that aerobic conditions resulted in similar titers compared to the oxygen-limited conditions and the cells were able to consume glucose faster, leading to higher productivities (**Fig. S4C**).”

4. The symbols and abbreviations used in the figures should be explained at their first mention in the figure. For example, the abbreviations in Fig.1 (SUC2, HXK, etc.) should be explained.

We have expanded the caption for **Fig. 1B** to explain the abbreviations for various genes and genetic modifications.

“Deleted genes are marked in blue. Overexpressed genes are marked in red. G6P glucose-6-phosphate, GADP glyceraldehyde-3-phosphate, 1,3-BPG 1,3-bisphosphoglycerate, DHAP dihydroxyacetone phosphate, DHA dihydroxyacetone, G3P glycerol-3-phosphate, OAA oxaloacetate, HXK hexokinase, NDE external NADH dehydrogenase, GPD glycerol-3-phosphate dehydrogenase, PDC pyruvate decarboxylase, PYC pyruvate carboxylase, MDH malate dehydrogenase, FUMR fumarase, FRD fumarate reductase, MAE1 dicarboxylic acid transporter, SUC2 invertase.”

5. From Fig. S7, different initial glucose concentration lead to different SA yield. How do you decide glucose concentration of 50 g/L? Did you try other concentration? Similarly, how do you decide the substrate ratio of 50 g/L glucose and 20 g/L glycerol? Did you try other ratio?

For **Fig. S7**, the yields obtained when 50 g/L or 70 g/L of glucose were used were similar at 0.50-0.51 g/g. Only when 50 g/L of glucose and 20 g/L of glycerol were used, the yield was higher because more reducing equivalent NADH was produced from glycerol. For fermentations using shake flasks, initial glucose concentration of 50 g/L is commonly used in several metabolic

engineering studies. For the glycerol concentration, we optimized its value relative to the glucose concentration of 50 g/L. We have added the following results in the revised manuscript:

Lines 194-200: “We also compared the SA production in strain g3473 Δ /PaGDH-DAK using 50 g/L of glucose and different glycerol concentrations of 10 g/L, 20 g/L, and 30 g/L (**Fig. S8**). As expected, SA titer increased as the initial amount of glycerol increased; thus, a comparison between titers was not meaningful. Also, the yield was lowest when 10 g/L of glycerol were used. No significant difference in yields was observed when 20 g/L and 30 g/L of glycerol were used; however, the productivity was highest when 20 g/L of glycerol were used. Thus, we considered 20 g/L of glycerol to be the optimal concentration to use with 50 g/L of glucose for the SA production.”

6. The functional unit and system boundary of LCA should be defined.

We thank the reviewer for correctly pointing out that we did not explicitly mention the functional unit of our LCA, which was 1 kg of succinic acid in the product stream. We have accordingly modified the section titled “Techno-economic analysis and life cycle assessment” in Materials and Methods (see below). Further, while the scope of our LCA was mentioned (cradle-to-grave for GWP₁₀₀; cradle-to-gate for FEC), the reviewer correctly pointed out that we did not explicitly mention the system boundary, which was the entire biorefinery including feedstock juicing, fermentation, separation, wastewater resource recovery, a boiler, and a turbogenerator. Accordingly, we have added a new system boundary figure (**Fig. S17**) in the Supporting Information and modified the text in the section titled “Techno-economic analysis and life cycle assessment” to reflect this (see below).

Lines 642-645: “The MPSP was estimated in 2016\$. The functional unit of the TEA and LCA was 1 kg of succinic acid in the product stream. The scope of the LCA was cradle-to-grave for GWP₁₀₀ (assuming the end-of-life degradation of the product succinic acid entirely to CO₂) and cradle-to-gate for FEC. The system boundary of the LCA is depicted in **Fig. S17**.”

New **Fig. S17** in the Supporting Information:

7. Did you analyze the purity of your SA crystal, and did you consider purity when doing TEA? As the purity will influence the economic analysis in TEA.

Thank you for your comment. We did analyze the purity of the recovered SA crystal and considered this in the TEA. In the revised manuscript, the details of the process we used to determine the purity of SA crystals were incorporated in the Material and Methods section titled “SA crystallization, recovery, and purity.” Accordingly, the Results section titled “Fed-batch fermentations and scale-up” was modified. For TEA and LCA, we conservatively assumed the purity of succinic acid achieved was 85%. All TEA and LCA results (MPSP, GWP₁₀₀, and FEC) are reported with a functional unit of 1 kg of succinic acid in the product stream (i.e., weighted by purity; e.g., \$ per kg of succinic acid in the product stream).

Materials and Methods: SA crystallization, recovery, and purity

Lines 625-633: “The SA crystals recovered via crystallization in stage I and stage II were separated from the fermentation broth by simple filtration. From both crystallization stages, the recovered SA crystals were washed with cold Millipore water (4 °C) to eliminate any impurities from the crystal surface. Later, the washed SA crystals were dried at 50 °C for 24 hours for subsequent analysis. To measure the SA purity, a solution of dried crystals (stage I and II) of concentration 5 g/L (0.1 g in 20 mL) was made utilizing Millipore water. Later, the SA weight in the crystal was calculated by comparing the peak area against the analytical grade SA solution of same

concentration procured from Sigma-Aldrich. The analysis was carried out in duplicate and the purity was represented by equation (2).

$$\% \text{ purity} = \frac{\text{Weight of SA in crystal recovered}}{\text{Weight of crystal taken}} \times 100 \quad (2)''$$

Results: Fed-batch fermentations and scale-up

Lines 272-277: “Furthermore, the purities of SA crystals recovered in stage 1 and stage 2 were estimated to be 88.9% and 86.23%, respectively. The obtained result was in line with the findings from a previous study³⁰. From the results, it was evident that the crystallization of SA in high purity (>85%) from untreated fermentation broth was successful. However, further investigation is needed to eliminate the coloring impurities to recover SA crystal of commercial grade.”

Reviewer #3:

This manuscript studied metabolic engineering strategies, performed both fermentation experiments and TEA and LCA on biological conversion of sugar to succinic acid (SA) production pathway. The study is inclusive studies of both fermentation experiments and cost/sustainability analysis to guide R&D. In that regard, the manuscript is a comprehensive study of SA conversion pathways. My general comments are:

1. Several previous literatures have already demonstrated high yield and high titer. What is uniqueness and values of metabolic engineering practices here?

In our study, we used the acid-tolerant *I. orientalis* strain for succinic acid production. Indeed, succinic acid production has been achieved with high titers, yields, and productivities using various engineered microorganisms. However, bacteria were commonly used as the hosts, which grew best at neutral pH conditions. Since low pH conditions and accumulation of succinic acid in the undissociated form at low pH are toxic to bacterial growth, fermentations were performed at neutral pH conditions by addition of neutralizing agents, such as calcium carbonate. However, for a sustainable and economical SA production process, downstream separation and purification processes must be considered, in addition to just the fermentative performance (titer, yield, and productivity). After fermentation, strong acids, such as sulfuric acid, were needed to convert the salt form of succinic acid into protonated succinic acid with gypsum (CaSO₄) forming as a byproduct. The requirement of neutralizing agents and strong acids complicates the fermentation, downstream separation, and purification processes and increases the cost of the whole production process. These issues are further exacerbated as the scale of the fermentation becomes larger.

On the other hand, our engineered *I. orientalis* strains could produce more than 100 g/L of SA in sugar-based media at low pH (pH 3), significantly reducing the amount of neutralizing agents

needed during fermentation. To our knowledge, this is the first reported yeast strain that could achieve such production. Furthermore, succinic acid crystals could be directly crystallized from the fermentation broth without further acidification of the broth, enabling simple downstream separation and purification processes. The advantages of low pH fermentation using *I. orientalis* were further confirmed by our TEA/LCA, which showed that at the same yield, titer, and productivity combinations, the MPSP, GWP₁₀₀, and FEC values for low pH fermentation were lower compared to those for fermentation at neutral pH (**Fig. 4C–F** and **Fig. S14B,C**). The decrease in MPSP, GWP₁₀₀, and FEC values was due to the reduced base and acid requirements associated with low-pH fermentation

2. I understand glycerol and sucrose can improve overall yield and titer, but those two substrates were not discussed extensively for the readers to fully interpretate the supply chain constraints and impacts, my suspension is whether those substrates are only for high yield demonstration? If that is the case, is it practical?

Yes, glycerol was used to increase the SA yield. For TEA and LCA, we considered only clarified sugarcane juice—which contained sucrose, glucose, and fructose—as the main substrate. We have discussed the supply chain constraint and market price of sugarcane juice in the response to the comment on Page 10 below.

3. I do have some reservation on the uncertainty analysis. For instance, typical Monte Carlo analysis requires over 5,000 runs, so I am not sure whether 2,000 runs performed in this study would be adequate

We agree with the reviewer that it is not uncommon to run more simulations than 2,000 to achieve reproducible results. However, as mentioned in the “Techno-economic analysis and life cycle assessment” section of the Results, we have used the well-established Latin hypercube sampling method to generate samples from the input distributions for our uncertain parameters. It is well established that Latin hypercube sampling can reduce the number of simulations required to achieve reproducible results. To confirm that results were reproducible, we completed the analysis with 1,000, 2,000, 3,000, and 5,000 simulations and reported all of them in the Supporting Information. As shown below, results converged to within 0-1% for MPSP and GWP₁₀₀ across all percentiles (median, 5th, 10th, 25th, 75th, 90th, and 95th percentile), and to within 1-5% for FEC (for all except the 50th percentile for FEC, which was close to 0 MJ/kg) after 2,000 Monte Carlo simulations.

Results: Techno-economic analysis and life cycle assessment

Lines 290-291: “the distribution of results at alternative numbers of Monte Carlo simulations is reported in **Table S12**”

Supporting Information

Table S12

Percentile	MPSP [\$/kg]				GWP ₁₀₀ [kg CO ₂ -eq./kg]				FEC [MJ/kg]			
	1000	2000	3000	5000	1000	2000	3000	5000	1000	2000	3000	5000
5th	1.23	1.23	1.23	1.23	1.23	1.22	1.22	1.23	-7.02	-7.08	-7.17	-6.90
10th	1.26	1.26	1.26	1.26	1.35	1.32	1.32	1.33	-4.40	-5.54	-5.40	-5.41
25th	1.31	1.31	1.31	1.31	1.50	1.50	1.51	1.50	-2.56	-2.75	-2.64	-2.77
50th	1.37	1.37	1.37	1.37	1.70	1.71	1.70	1.70	0.145	0.265	0.207	0.140
75th	1.44	1.44	1.44	1.44	1.89	1.91	1.90	1.90	2.68	2.93	2.83	2.94
90th	1.50	1.50	1.50	1.50	2.06	2.07	2.08	2.07	5.03	5.26	5.21	5.15
95th	1.53	1.54	1.54	1.54	2.17	2.17	2.20	2.18	6.41	6.47	6.75	6.50

4. The main text called out tables and figures from supplementary information more frequently than other papers, the readers have to cross check the two documents almost all the time. What is the point of adding that valuable information into supplementary information then? Would it be more convenient just to combine into the main text?

Most of the figures in the supplementary information are fermentation profiles for individual engineered *I. orientalis* strains. For the main text, we prefer to summarize just the titers and productivities of these strains, instead of showing all the fermentation profiles. For these fermentation profiles, we recommend the readers to refer to the supplementary information.

Specific comments are also listed below:

- Title: “pipeline” is an interesting choice of word.

We appreciate your comment. Here we simulated a production pipeline that started accepted sugarcane as a feedstock, saccharified it to sugarcane juice (sucrose, glucose, and fructose), converted the sugars to SA using *I. orientalis*, and separated the fermentation broth to recover dried SA crystals.

5. Table 1: other than larger reactor demonstration, what are the unique contribution from this study on improving SA fermentation technologies? The fermentation performance from reference 65 seems more favorable. Also why 75L reactor resulted in a significant yield reduction?

The present scale-up work was the first demonstration at the pre-commercialization stage and a key enabler in commercialization. Before commercialization, several such tests are required for the steady and smooth transition of the production process into the market. Our study showed that an end-to-end manufacturing of SA was possible via the biological route without generating any pollutants at a lower cost compared to other published processes. Additionally, the key finding is that the crystallization of untreated broth is successful without generating any secondary molecule or pollutant. Yet, the apparent amount of SA remains miscible in the fermentation broth and requires more investigation and optimization. Additionally, the presence of other solutes, and

coloring impurities in the fermentation broth has been a major bottleneck to the crystallization process and can be attributed to the lower SA yield during crystallization.

For the indicated reference, Wang et al. employed a riboregulator switch system in *E. coli* to manufacture SA. The work reported SA yield and productivity of 0.91 g/g and 3.25 g/L/h respectively. However, the near-neutral pH fermentation was carried out at a 3 L scale with a working volume of 1.2 L. This near-neutral pH fermentation has been a bottleneck for downstream processing to recover SA. In this work, for the first-time, a low-pH fermentation (pH 3) and scale-up to produce SA at high titer, yield, and productivity in a yeast strain was demonstrated. As discussed above, our engineered *I. orientalis* strains simplified downstream separation and purification processes, enabling sustainable and economical biomanufacturing of SA that could be commercialized.

Lastly, the yield obtained from the batch fermentation in pilot-scale reactor (0.50 g/g glucose equivalent) was lower than the yields obtained from fed-batch fermentations using bench-top reactors using our engineered *I. orientalis* (0.63-0.65 g/g glucose equivalent) (**Table 1**). This is typically the case. For batch fermentations, the initial cell density is usually low, and cells need to adapt to a new environment and use the biomass for cell growth, leading to lower yield. On the other hand, for fed-batch fermentations, when the initial substrates are depleted, the culture has already reached higher cell density. As more substrates are added to the bioreactor, the cells will then be able to use most of the substrates for production instead of cell growth, leading to higher yield. As shown in **Fig. 3A**, for the first 72 hours of the batch fermentation using SC-URA medium, the OD₆₀₀ (cell density) increased rapidly, and the SA yield was 0.55 g/g glucose equivalent. For the fed-batch fermentation from 72 to 288 hours, the OD₆₀₀ values were similar, and the final yield was 0.63 g/g glucose equivalent, higher than the yield from the batch fermentation. For fermentations using sugarcane juice, the OD₆₀₀ values were not recorded since the sugarcane juice contained a lot of solid materials, which would not give accurate measurements of cell densities.

Similarly, for our fermentations using sugarcane juice in bench-top reactors, the yield obtained from batch fermentation was only 0.47 g/g glucose equivalent (comparable to 0.5 g/g glucose equivalent obtained in pilot-scale fermentation), and while the yield obtained from the fed-batch fermentation was higher at 0.63 g/g glucose equivalent. We anticipate that we would be able to obtain higher yield at pilot scale for fed-batch fermentation, which was not performed due to budget, materials, and volume considerations.

- Page 9, “we designed and simulated biorefineries across the fermentation performance landscape (i.e., 2,500 yield-titer combinations each across a range of productivities for both neutral and low-pH fermentation) to set and prioritize targets for further improvements to financial viability and environmental sustainability”. The definition of variable distribution should be based on experimental data or warranted experiences, not with a design. This will directly impact the credibility of the cost distribution.

We fully agree with the reviewer that any scenarios representing the current state of the technology should be based on experimental data. Indeed, the baseline values and uncertainty distributions for fermentation titer, yield, and productivity (as well as other parameters) in our *lab batch scenario*, *lab fed-batch scenario*, and *pilot batch scenario* are all based on the experimental data from this work (see Table S6 in the SI for detailed reporting of all assumed baseline values and uncertainty distributions for these and other parameters). The analyses across 2,500 fermentation yield-titer combinations were performed to prioritize future fermentation advancements; i.e., to demonstrate the implications of improvements to yield and titer on system-wide sustainability. We agree that this could be made clearer, and have modified the sentence highlighted by the reviewer.

Results: Techno-economic analysis and life cycle assessment

Lines 297-300: “Finally, to set and prioritize targets for further improvements to financial viability and environmental sustainability, we designed and simulated biorefineries across the potential fermentation performance landscape (i.e., 2,500 yield-titer combinations each across a range of productivities for both neutral and low-pH fermentation) ~~to set and prioritize targets for further improvements to financial viability and environmental sustainability.~~”

- Page 10, “We engineered the strains by deletion of byproduct pathways, transport engineering to enable SA export and limit SA import, and expansion of the substrate scope to allow utilization of glycerol and sucrose.” Supply constraints and price impacts are not discussed in the TEA or LCA, which should be critical to address the questions on why including these two substrates to SA fermentation/production.

Material supply and prices are key considerations in TEA. This is why we have attributed uncertainty to the prices of all material and utility inputs and to the price and availability of our feedstock (sugarcane, which contains sucrose, glucose, and fructose; see Table S6 in the SI for baseline values and distributions of all parameters). Indeed, the uncertainties in feedstock price and capacity were found to be important drivers of the uncertainty in MPSP (with Spearman’s rank order coefficients of 0.39 and -0.31 with respect to MPSP, respectively; see Table S9 of the SI for all Spearman’s rank order correlation coefficients) but, as discussed in the manuscript, the uncertainty in fermentation yield was found to be much more significant to MPSP than other parameters (with a Spearman’s rank correlation coefficient of -0.60 with respect to MPSP). However, we agree with the reviewer that the significance of feedstock price and availability could be better noted in the main manuscript, and have accordingly made modifications as shown below.

Results: Techno-economic analysis and life cycle assessment

Lines 315-323: “Across the 28 parameters to which uncertainty was attributed for the *pilot batch scenario*, we found MPSP was most sensitive to fermentation SA yield (Spearman’s ρ of -0.60; all uncertainty distributions are listed in **Table S6** and Spearman’s ρ values for all parameters are reported in **Table S9**). MPSP was also significantly sensitive to feed sugarcane unit price

(Spearman's ρ of 0.39), plant uptime (-0.38), the plant's capacity for feed sugarcane (-0.31), and fermentation SA titer (-0.30). GWP₁₀₀ was most sensitive to the boiler efficiency (Spearman's ρ of -0.63). GWP₁₀₀ was also significantly sensitive to fermentation SA titer and yield, with Spearman's ρ values of -0.62 and 0.32, respectively. FEC was most sensitive to fermentation SA yield (Spearman's ρ of 0.63), and also sensitive to the boiler efficiency (-0.56) and fermentation SA titer (-0.49)."

Discussion

Lines 415-419: "From the sensitivity analysis we performed for the *pilot batch* scenario, we found MPSP to be most sensitive to fermentation SA yield and relatively less sensitive to fermentation SA titer. However, we found GWP₁₀₀ and FEC to be more sensitive to fermentation SA titer than to yield, demonstrating the importance of these parameters in achieving a financially viable and environmentally sustainable full-scale process."

- Page 12, "the baseline feedstock sugarcane cost assumed in our work was \$49.3/dry metric ton)." What is the basis/credible reference to assume this feedstock cost? The cost number seems low to me.

The baseline value for feedstock sugarcane price was \$49.3/dry metric ton in 2016\$, or \$34.5/wet metric ton; this value was based on Huang et al., 2016, who reported a value of \$35/wet metric ton (a full list of assumed baseline values, uncertainty distributions and their sources is available in Table S6). Because we wanted to characterize the implications of uncertainty in this price, we attributed a large uncertainty of 20% below and above the baseline value as a uniform distribution. Indeed, we found the uncertainty in feedstock price to be a significant driver of uncertainty in MPSP, albeit not as significant as the uncertainty in the fermentation yield. We have therefore modified the manuscript text to further highlight the significance of feedstock price and availability (as discussed in our response to the previous comment). We have also made modifications to our discussion section to further highlight the sources of the baseline values and uncertainty distributions. Further, we have modified **Table S6** to include Huang et al., 2016 (the original study cited by Cortes-Pena et al., 2020) as a reference.

Results: Techno-economic analysis and life cycle assessment

Lines 315-323: "Across the 28 parameters to which uncertainty was attributed for the *pilot batch scenario*, we found MPSP was most sensitive to fermentation SA yield (Spearman's ρ of -0.60; all uncertainty distributions are listed in **Table S6** and Spearman's ρ values for all parameters are reported in **Table S9**). MPSP was also significantly sensitive to feed sugarcane unit price (Spearman's ρ of 0.39), plant uptime (-0.38), the plant's capacity for feed sugarcane (-0.31), and fermentation SA titer (-0.30). GWP₁₀₀ was most sensitive to the boiler efficiency (Spearman's ρ of -0.63). GWP₁₀₀ was also significantly sensitive to fermentation SA titer and yield, with Spearman's ρ values of -0.62 and 0.32, respectively. FEC was most sensitive to fermentation SA

yield (Spearman’s ρ of 0.63), and also sensitive to the boiler efficiency (-0.56) and fermentation SA titer (-0.49).”

Discussion

Lines 400-404: “and the baseline values and uncertainty distributions assumed in our work for all parameters are listed in **Table S6**.”

Supporting Information

Table S6. Details of parameters included in the uncertainty analysis. For neutral fermentation, the base required: succinic acid produced is assumed to be constant at 2 mol-OH-eq./mol, and the sulfuric acid requirement for downstream acidulation is assumed to be 2-mol-H⁺-eq./mol-succinic-acid.

Feedstock unit price	\$/wet-kg	3.45E-02	Triangular	2.76E-02	3.45E-02	4.14E-02	baseline based on Ref. 27; bounds are $\pm 20\%$ of baseline
-----------	----------	------------	----------	----------	----------	--

New reference 27 in the Supporting Information

27. Huang H., Long S. & Singh V. Techno-economic analysis of biodiesel and ethanol co-production from lipid-producing sugarcane. *Biofuels. Bioprod. Bioref.* **10**, 299– 315 (2016).

- Page 12, “The second of the two studies had different fermentation performance assumptions (yield of 0.96 g/g, titer of 55.8 g/L, and productivity of 0.77 g/L/h) compared to the fermentation performance achieved at the pilot scale in our work (0.473 g/g, 63.1 g/L, and 0.657 g/L/h, respectively).” Surprised to see such a low yield. Any explanation on the yield differences between this study and the study with 0.96 g/g yield.

For their biorefinery simulation, Dogbe et al. used the fermentative performance of an *E. coli* strain engineered by Chan et al. (yield of 0.96 g/g, titer of 55.8 g/L, and productivity of 0.77 g/L/h)^{1,2}. Yeasts, in general, cannot biologically achieve as high SA yield compared to bacteria due to the lack of cytosolic NADH, which is the cofactor required in the reductive TCA pathway. For yeast species, such as *I. orientalis*, all cytosolic NADH is produced from glycolysis at yield of 2 NADH molecules per 1 glucose molecule. Bacterial species have no cellular compartments and thus can use the NADH produced from the TCA cycle, in addition to NADH produced from glycolysis, for SA production; on the other hand, NADH produced from the TCA cycle in yeasts is confined in the mitochondrion and cannot be transported to the cytosol. Thus, SA yields obtained from yeasts are lower than yields obtained by bacteria.

References:

1. Dogbe, E.S., Mandegari, M. & Görgens, J.F. Revitalizing the sugarcane industry by adding value to A-molasses in biorefineries. *Biofuels. Bioprod. Bioref.* **14**, 1089–1104 (2020).

2. Chan, S., Kanchanatawee, S. & Jantama, K. Production of succinic acid from sucrose and sugarcane molasses by metabolically engineered *Escherichia coli*. *Bioresour. Technol.* **103(1)**, 329-336 (2012).

• Page 13, “...while SA yield is more critical to the economics of the biorefinery than titer at the current state of the technology, continued improvements to titer represent the greatest opportunity to reduce environmental impacts.” I enjoyed this discussion, but this is a bit qualitative. IF you could add quantitative discussion on at what specific level of yield the yield becoming critical to cost and GHG, that would be helpful to the readers.

We had included a quantitative discussion on the relative significance of fermentation titer and yield on the MPSP, GWP₁₀₀, and FEC in the Supporting Information. However, we agree with the reviewer that this should be better highlighted to the reader, and we have therefore moved this discussion to the main manuscript as shown by the modifications below.

Discussion

Lines 438-451: “For example, at the *pilot batch scenario* fermentation yield (36.1% of the theoretical maximum) and titer (63.1 g/L), improving yield by 3.6% (a 10% relative increase) would decrease the MPSP by \$0.06/kg, while improving titer by 6.3 g/L (a 10% relative increase) would decrease the MPSP by \$0.03/kg. However, improvements to titer have much greater potential benefits to GWP₁₀₀ and FEC, as increasing titer would decrease heating and cooling utility demands while increasing yield would increase these environmental impacts due to higher electricity consumption: at a fixed titer, an increased succinic acid yield on sugars results in a larger fermentation vessel size required, which necessitates higher mixing power requirements (which increase with fermentation vessel size). In the baseline case for the *pilot batch scenario*, no natural gas is purchased to satisfy the heating utility demand; natural gas is purchased solely for the gas-fired dryer that removes moisture from crystallized succinic acid. In fact, enough steam is produced to completely satisfy the heating and power utility demands and produce excess electricity (using a turbogenerator) to be sold back to the grid and displace the GWP₁₀₀ and FEC impacts associated with grid electricity (**Fig. 4E–F** and **Fig. S14B–C**; a detailed description of the simulated co-heat and power generation configuration is available in a previous study⁵⁶). ~~the relative significance of yield and titer on MPSP, GWP₁₀₀, and FEC is discussed further in the SI~~”

Supporting Information

~~Discussion on relative significance of yield and titer on biorefinery economics and environmental impacts~~

~~At the baseline fermentation yield-titer combinations for *laboratory batch*, *laboratory fed-batch*, and *pilot batch scenarios*, improvements to yield had a greater benefit for MPSP than comparable relative improvements to titer. For example, at the *pilot batch* fermentation yield (36.1% of the theoretical maximum) and titer (63.1 g/L), improving yield by 3.6% (a 10% relative increase) would decrease the MPSP by \$0.06/kg, while improving titer by 6.3 g/L (a 10% relative increase) would decrease the MPSP by \$0.03/kg. However, improvements to titer have much greater potential benefits to GWP₁₀₀ and FEC, as increasing titer would decrease heating and cooling utility demands while increasing yield would increase these environmental impacts due to higher electricity consumption: at a fixed titer, an increased succinic acid yield on sugars results in a larger fermentation vessel, which necessitates higher mixing power requirements (which increase with fermentation vessel size). In the baseline case for the *pilot batch scenario*, no natural gas is purchased to satisfy the heating utility demand; natural gas is purchased solely for the gas-fired dryer that removes moisture from crystallized succinic acid. In fact, enough steam is produced to completely satisfy the heating and power utility demands and produce excess electricity (using a turbogenerator) to be sold back to the grid and displace the GWP₁₀₀ and FEC impacts associated with grid electricity (a detailed description of the simulated co-heat and power generation configuration is available in Bhagwat et al., 2021⁴).~~

• Page 14, “Crude glycerol, as a waste, is also a no-cost or low-cost substrate, ...” I totally disagree with this assumption. Nothing is free.

We have removed this part from the manuscript. For TEA/LCA, we performed simulations using only sugarcane as the feedstock and clarified sugarcane juice as the fermentation substrate.

Reviewers' Comments:

Reviewer #1:

Remarks to the Author:

The manuscript has been carefully revised. All the problems and comments has been answered and solved. So, the manuscript might be accepted for publication now.

Reviewer #3:

Remarks to the Author:

I believe the authors have adequately addressed all my comments.

We thank all the reviewers again for their thoughtful comments. We are pleased that they were satisfactory with our revisions. For the additional editorial requests, we have fully addressed them as indicated in the author checklist. All the corresponding changes are highlighted in red in the revised manuscript.

Reviewer #1 (Remarks to the Author):

The manuscript has been carefully revised. All the problems and comments has been answered and solved. So, the manuscript might be accepted for publication now.

We wholeheartedly thank the reviewer for his/her help in improving the quality of our manuscript.

Reviewer #3 (Remarks to the Author):

I believe the authors have adequately addressed all my comments.

We wholeheartedly thank the reviewer for his/her help in improving the quality of our manuscript.